


**Modeling a tropical-like cyclone in the Mediterranean Sea**
**under present and warmer climate**

Shunya Koseki[1], Priscilla A. Mooney[2], William Cabos[3],
Miguel Ángel Gaertner[4], Alba de la Vara[4], Juan Jesus González Alemán[5]

*1: Geophysical Institute, University of Bergen / Bjerknes Centre for Climate Research, Bergen, NORWAY*
*2: NORCE Norwegian Research Centre AS / Bjerknes Centre for Climate Research, Bergen, NORWAY*
*3: Departamento de Ciencias Fisica, Universidad de Alcalá, Alcalá de Henares, SPAIN*
*4: Facultad de Ciencias del Medio Ambiente, Universidad de Castilla-La Mancha, Toledo, SPAIN*
*5: Departamento de Física de la Tierra y Astrofísica, Universidad Complutense de Madrid, Madrid, SPAIN*

Correspondence to Shunya Koseki
Address: Allégate 70, 5007, Bergen, Norway
Email: Shunya.Koseki@gfi.uib.no



## Abstract

This study focuses on a single Mediterranean hurricane (hearafter medicane), to investigate the medicane response to global warming during the middle of the 21$^{st}$ century and assess the contradictory effects of a warmer ocean and a warmer atmosphere on its development. Our investigation uses the state-of-the-art regional climate model WRF with the optimum combination of physical parameterizations based on a sensitivity assessment study. Results show that our model setup can reproduce a realistic cyclone track and the transition from initial disturbance to tropical-like cyclone with a deep warm core although the transition is earlier than for the observed medicane. To investigate the response of the medicane to future climate change, a pseudo global warming (PGW) approach has been used. This approach adds the projected change of atmospheric and ocean variables obtained by an ensemble of CMIP5 models to the boundary conditions for the regional climate model. A PGW simulation where all variables (PGW$_{ALL}$) are incremented shows that most of the medicane characteristics moderately intensify, e.g., surface wind speed, uptake of water vapour and precipitation. However the maximum depression of sea level pressure (SLP) is almost identical with that under present climate conditions. Two additional PGW simulations were undertaken; One simulation adds the projected change in sea surface and skin temperature only (PGW$_{SST}$) while the second simulation adds the PGW changes to only atmospheric variables (PGW$_{ATMS}$) i.e. we use present time sea surface temperatures. These simulations show opposite effects on the medicane. In PGW$_{SST}$, the medicane is reinforced more vigorously than PGW$_{ALL}$: much deeper SLP depression, stronger surface wind, and more intense evaporation and precipitation. In contrast, the medicane in PGW$_{ATMS}$ weakens considerably (SLP, surface wind and rainfall decrease) still converts into a tropical-like cyclone with a deep warm core. This difference can be explained by an increased water vapour driven by the warmer ocean surface (favourable for cumulus convection) and the warmer and drier atmosphere in PGW$_{ATMS}$ tends to inhibit condensation (unfavourable for cumulus convection). As a result of these counteracting effects of warmer ocean and atmosphere, the medicane is enhanced only modestly by global warming.



## 1. Introduction

It is well known that severe cyclonic storms occur in the Mediterranean Sea, in particular, from September to March (e.g., Cavicchia et al., 2013). These cyclonic storms occasionally develop into meso-scale cyclones with a cloud-free "eye" around the cyclone centre, which is also a common typical feature of tropical cyclones. They generate vigorous precipitation and intense wind, resulting in disastrous damages on regional economies and infrastructure over the coastal areas in the Mediterranean nations (e.g., Bakkensen, 2017). These tropical-like cyclones are called Mediterranean hurricanes or medicanes even though their lifetime as tropical-like cyclones is shorter than most tropical cyclones. In general, the initiation and development of a tropical cyclone requires certain physical and dynamical conditions (e.g., Camargo et al., 2007). One of these is warm sea surface temperature (SST) like the warm pool in the tropical western Pacific Ocean. However, the autumn and winter SST in the Mediterranea Sea varies from around 18°C to, at highest, 23°C in the current climate (e.g., Shaltout and Omstedt, 2014; Fig. 2a), which is much lower than tropical SST and the generation of tropical cyclone is quite rare over such cold SST (cf. Pacific and Atlantic cold tongue, e.g., Jin 1996; Caniaux et al., 2011) even in the tropics.

The mechanism of medicane development has been investigated in previous studies. Similar to tropical cyclones, an initial atmospheric disturbance is essential for medicane genesis. In many cases, medicanes arise from cut-off lows isolated from the extratropical jet stream in the upper troposphere (e.g., Emanuel, 2005; Fita et al., 2006; Chaboureau, et al., 2012). Such a cut-off low and a potential vorticity anomaly are pre-conditional for medicane initiation. This triggers deep cumulus convection resulting in the formation of a deep warm core and consequently, a tropical transition of the initial perturbation occurs (Mazza et al., 2017; Fita and Flaounas, 2018). Miglietta and Rotunno (2019) showed the importance of air-sea interactions for the development of two medicanes. These discrepancies in the literature may arise from the dependency of the various studies onindividual case studies. The surface wind associated with the cyclonic circulation induces the turbulent heat fluxes effectively and cumulus convection and diabatic heating are enhanced (wind-induced surface heat exchange, WISHE, e.g., Emanuel, 1986).

In both mechanisms of medicane development and tropical transition, diabatic heating due to condensation plays a vital role. As such understanding the response of medicane features to anthropogenic global climate change is important for mitigating future risks associated with these natural hazards. According to Shaltout and Omdtedt (2014), the Mediterranean SST is expected to increase by 2.6°C per century. This warming in the ocean can be a potential source of enhanced moisture to the atmosphere. In fact, significant changes in medicanes e.g., frequency and intensity associated with global warming have been reported in previous studies. Cavicchia (2014) showed that whereas the frequency of medicanes tends to decrease, medicanes can moderately intensify



based on a climate projection with a global coupled model. Tous et al. (2016) also suggested similar
future changes in frequency and intensity of medicanes. Their study also revealed that the location
of medicane formation is expected change (more frequent over the Gulf of Lion-Genova and South
of Sicily). González-Alemán et al. (2019) concluded that associated with medicanes intensification,
the structure of tropical-like cyclones is more robust and their life-time as tropical-like cyclones is
longer-lasting compared to medicanes under current climate This, consequently, leads to more
hazardous situations in the projected future.
Most of the aforementioned studies on the future climate of medicanes are based on results
obtained from global coupled models (CGCMs). However, in long climatic simulations perfomed
with CGCMs, the typical grid spacing varies between 100km and 25km at best. Even simulations at
25km are still insufficient to resolve the fine-scale structure of medicanes such as the cyclone core
and associated rain bands; the intensity of medicanes is underestimated in several coupled regional
models (Gaertner et al. 2018). Therefore, it is most likely that CGCMs also underestimate future
changes in medicanes. One possible solution to this problem is to dynamically downscale the global
models with a regional climate model (RCM) at finer resolutions. Alternatively, a pseudo global
warming method (PGW, e.g., Schär et al., 1996; Rasmussen et al., 2011; Parker et al., 2018;
Mooney et al., in review) can be used to assess more explicitly the impacts of future climate change
on medicanes. PGW is an advantageous method to characterize a given medicane in current and
future climate by imposing the future changes in atmospheric and ocean variables estimated by
CGCMs to boundary conditions of a high-resolution RCM (see details in Section 2). This approach
permits a more direct assessment of impacts of future climate change on an extreme weather event
(e.g., Parker et al., 2018). Additionally, the PGW method enables investigations of the relative roles
of a warmer atmosphere and a warmer ocean in the response of medicanes to climate change.
In this study, based on a PGW framework, we investigate the impacts of global warming on
the development and intensity of medicane Rolf (Miglietta et al., 2013; Ricchi et al., 2017; Dafis et
al., 2018). Rolf occurred from 6[th] to 9[th] November in 2011 and affected the Balearic Islands, Italy
and south France due to longer persistence of tropical-cyclonic features. Rolf is the first tropical
cyclone officially monitored by the National Oceanic and Atmospheric Administration (NOAA).
Since Rolf was a highly destructive medicane for coastal communities in many Mediterranean
countries, it is important to assess how these types of medicanes will respond to climate change in
near future. We perform additional idealised experiments in which only the atmosphere or the
ocean, respectively, experience global warming to elucidate the roles of a warmer atmosphere and a
warmer ocean on the medicane. This study is structured as follows. In Section 2, details of the
reanalysis data, RCM and experimental designs are provided. An evaluation and assessment of the
simulation of medicane Rolf under current climate conditions with respect to a state-of-the-art


reanalysis is presented in Section 3. The results of the PGW experiments are given in Section 4.
Additionally, we will analyze the possible future changes of the medicane. A more insightful
discussion on the competing roles of a warmer atmosphere and ocean in the medicane, respectively,
will be examined in Section 5. Finally, the concluding remarks of this study will be provided in
Section 6.

**2. Data, Model, and Methodology**
In this study, ERA5 reanalysis data (Copernicus Climate Change Service, 2017) is used to
benchmark the medicane Rolf simulation. ERA5 is a state-of-the-art reanalysis system with a high
spatio-temporal resolution (0.25°×0.25° and 1 hourly). The trajectory of Rolf computed from ERA5
data is regarded as the best medicane track in this study. Additionally, we use observational data of
the cyclone track produced by the US National Oceanic and Atmospheric Administration (NOAA).
This data is available only from 12UTC, 7 to 12UTC, 9 of November in 2011
(https://www.ssd.noaa.gov/PS/TROP/DATA/2011/tdata/med/01M.html).

*2.1 WRF Simulation of Rolf under present climate*
Simulations of Rolf are performed with the Weather Research and Forecasting (WRF,
Skamarock et al., 2008) model version 3.9.1. The experimental domains consist of two nested
domains as shown in Fig.1: the first domain has a 15km mesh covering (-14.3386°E, 34.3386°E)
and (30.3248°N, 52.6961°N) and the second one has a 5km mesh covering (-3.9688°E, 12.4400°E)
and (35.3447°N, 47.5810°N) with 52 vertical layers, respectively. In both domains, a cumulus
convection scheme is switched on. Previous studies (e.g., Migliette et al., 2015; Ricchi et al., 2017;
Mooney et al., 2018) have shown that simulated medicanes and tropical cyclones are highly
sensitive to different combinations of physical schemes. Therefore, we investigated different
plausible combinations of physical schemes for Rolf in a 10km-mesh forced by ERA-Interim (Dee
et al., 2011) reanalysis data (0.75°×0.75°, 6 hourly). In this assessment, we used 27 combinations of
physical schemes among cumulus convection (Kain-Fritsch (Kain 2004), Betts-Miller-Janjíc
(Janjíc, 1994), Tiedtke (Tidikte, 1989; Zhang et al., 2011)), microphysics (WSM5 (Hong et al.,
2004), WSM6 (Hong and Lim, 2006), Thompson (Thompson et al., 2008)) and planetary boundary
layer (Yonsei Univeristy (Hong et al., 2006), Mellor-Yamada-Janjíc (Janjíc, 1994), Mellor-
Yamada-Nakanishi-Niino (MYNN; Nakaish and Niino, 2006; Nakanishi and Niino, 2009; Olson et
al., 2019)). The assessment simulations showed that most simulations of Rolf with Tiedtke
convection scheme make a landfall around southern France successfully and all simulated
medicanes with Kain-Fritsch and Betts-Miller-Janjíc convection schemes make an incorrect landfall
over the Sardinia Island or decay over the Mediterranean Sea without landfall (not shown). Based



on the results of this assessment, we select one of the best combinations: Tiedtke, Thompson,
MYNN. These three physical schemes are combined with the longwave and shortwave radiative
schemes of the Rapid Radiative Transfer Model (Mlawer et al., 1997) and NOAH 4-layer land
surface model (Chen and Dudhia, 2001a, b). Initialization and lateral boundary conditions are taken
from ERA-Interim 6-hourly reanalysis data ($0.75° \times 0.75°$). The lower boundary condition of sea
surface temperature (SST) is obtained from daily OISST data with $0.25° \times 0.25°$ horizontal
resolution. The simulations are integrated from 0000UTC on 05-Nov-2011 to 2300UTC on 10-Nov-
2011. These simulations are referred to PRS hereafter. ERA-Interim is selected as the driving data
for our WRF simulations to maintain consistency between the spatial resolutions of the PGW delta
calculated from the CMIP5 ensemble and the reanalysis data used for the initial and boundary
conditions (in particular for atmospheric variables). We also investigated the representation of the
medicane Rolf in ERA-Interim and found a cyclone track similar to ERA5 (see Section 3 for
details).

*2.2 WRF simulation of Rolf under warmer climate*

To investigate how future global climate change influences the medicane, a pseudo global

warming (PGW, e.g., Parker et al., 2018; Mooney et al., in review) experiment is employed. In the
PGW framework, boundary conditions of WRF are perturbed by the monthly-mean values of global
climate change ($\Delta$). This is estimated by simulations of climate projections from CGCMs. In other
words, we can simulate the medicane Rolf under a virtually warmed climate. In this study, we
obtain the PGW $\Delta$ from the ensemble mean (see Table 1) of 19 simulations used in the Coupled
Model Inter-comparison Project 5 (CMIP5, Taylor et al., 2012) between 2036-2065 and 1976-2005.
The PGW $\Delta$ contains perturbed values for zonal and meridional winds, temperature, relative
humidity, geo-potential, SLP, SST, and skin temperature. The new boundary conditions including
the global warming can be expressed as,

$$BC_{PGW} = BC_{PRS} + \Delta \ , \ \text{where}$$

$$\Delta = CMIP5_{2036-2065} - CMIP5_{2005-1976} \quad (1),$$

here, BC represents any of the variables used as boundary conditions for WRF.

In the PGW experiment, we perform three different simulations: (1) PGW $\Delta$ is added to all

the values of boundary conditions ($PGW_{ALL}$), (2) added only to SST and skin temperature
($PGW_{SST}$), and (3) added only to the atmospheric variables ($PGW_{ATMS}$). This enables an
investigation of the relative roles of projected future changes in the atmosphere and ocean in the
development and modification of the medicane. Other experimental configurations of PGWs are the



same as those in PRS (see section 2.1). Figure 2 provides the PGW Δ for SSTs and a vertical profile
of atmospheric temperature and relative humidity (averaged over the 5km-mesh domain in Fig. 1)
for the PGW experiments in this study. In the Mediterranean Sea, the SST increases approximately
2°C in Fig. 2b (also shown by Somot et al., 2006). The troposphere is entirely warming by 2 to 3°C
in Fig. 2c. In contrast, projections of the relative humidity in the troposphere tend to be reduced
under global warming. These projected thermodynamical responses to global warming can lead the
Mediterranean climate to be warmer and drier (e.g., Giorgi and Lionello, 2008). To our knowledge,
the present study is the first investigation to employ the PGW method to a tropical-like cyclone in
the Mediterranean Sea.

*2.3 Estimation of cyclone phase*

For a trajectory of observed and simulated medicanes, the minimum SLP is tracked from
00UTC-06-Nov-2011 until 12UTC-09-Nov-2011. If the medicane makes a landfall before 12UTC-
09-Nov-2011, the tracking is ceased. One remarkable characteristic of medicanes is the cyclonic
system transitions from extratropical to tropical (e.g., Gaertner et al., 2018). Hart (2003) proposes
an objective measurement of cyclone phase space defined as,

$$\left.\frac{\partial(\Delta Z)}{\partial \ln p}\right|_{900\,\text{hPa}}^{600\,\text{hPa}} = -\left|V_T^L\right| \ (2) \ \text{and} \ \left.\frac{\partial(\Delta Z)}{\partial \ln p}\right|_{600\,\text{hPa}}^{300\,\text{hPa}} = -\left|V_T^U\right| \ (3)$$

where,
$$\Delta Z = Z_{max} - Z_{min} \ (4).$$

$Z_{max}$ and $Z_{min}$ denote the maximum and minimum geopotential height at a pressure level
within 2.5° (for ERA5) and 250km (for WRF simulations) radius around the medicane centre. The
upper- and lower- tropospheric thermal wind relation is estimated by equations (2) and (3),
respectively. As shown by Hart (2003), in the extratropical phase, the cyclone has deep cold core
and the values of (2) and (3) are negative. In addition, the tropical cyclone has a deep warm core
with positive values for (2) and (3). In this study, the thermal wind relation is estimated every 50
hPa from 900 to 300 hPa and the cyclone phase indices of (2) and (3) are defined as mean of the
values every 50hPa between 600 and 300 hP and between 600 and 900 hPa.

**3. Simulation of Medicane Rolf under present climate**
In this section we examine the results of PRS to assess ability of our WRF setup to simulate
Rolf. Figure 3a gives a best track of Rolf estimated by ERA5. The best track is based on the




minimum value of SLP every 3 hour. At the beginning (0000UTC, 6-Nov), the cyclone centre was
identified around 5°E and 42°N in the Mediterranean Sea. This location of the cyclone centre is
associated with a cut-off low originated from the north Atlantic Ocean (not shown). The cyclone
moves southward and crosses the Balearic Islands from 1200UTC, 06-Nov to 0000UTC, 07-Nov.
Afterward, the cyclone shifts its direction to the east and north. According to previous studies (e.g.,
Dafis et al., 2018), Rolf reaches its peaks of deep cumulus convection from 0000 to 0300UTC on
08-Nov over the Mediterranean Sea. At 0000UTC on 09-Nov, Rolf approaches southern France and
part of it makes a landfall. After landfall, Rolf's intensity decays as it shifts northwestward and
disappears. The cyclone track of ERA5 is in good agreement with the NOAA observations. Even
though the location of Rolf after 0000UTC on 09-Nov is somewhat different, ERA5 observes the
westward progression of Rolf after 0000UTC on 09-Nov. The cyclone track of PRS is given in Fig.
3b. In general, the PRS simulation reproduces the observed cyclone track from the beginning to the
end of its life cycle. While the cyclone passes slightly more north of the Balearic Islands (landfall
over Ibiza), the cyclone moves northward at 0000UTC on 07-Nov and approaches southern France
at 00-09-Nov. In ERA5, Rolf shifts westward after the partial landfall on southern France, but the
PRS-simulated Rolf progresses northward and makes a complete landfall over southern France and
disappears afterward.

The phase shift of the cyclone of ERA5 and PRS is shown in Figure 4 as cyclone phase

space defined by Hart (2003) (see the details of definition in section 2.3). In ERA5, the cyclone has
a deep cold core at 0000UTC on 06-Nov (Figs. 4a and 4b), which is one of the characteristics of
extratropical cyclones. As time elapses, the cyclone transitions from a deep cold core to a shallow
warm core. In particular, the upper troposphere loses the feature of cold core during 7 of November
(Fig. 4a). After 1200UTC on 07-Nov, the warm core develops more vertically and consequently the
cyclone has a deep warm core structure at 0000UTC on 08-Nov. Approaching southern France at
2100UTC on 08-Nov (Fig. 3a), the upper troposphere shifts to a weak cold core indicating the
decay of the tropical-like cyclone. In PRS, at the beginning, the cyclone already develops with a
shallow warm core even though the upper troposphere has a strong cold core (Figs. 4c and d). This
structure of shallow warm core continues until 12-06-Nov and the cyclone forms a deep warm core
at 00-07-Nov (Fig. 4c and 4d), which is earlier than ERA5 (Figs. 4a and b). However, there is a still
cold core at 300 hPa (Figs. 4c) and at 00-08-Nov the simulated cyclone forms a completely deep
structure with strong warm core in the lower troposphere (Figs. 4c and d). This timing is almost
consistent with ERA5 even though the warm core is stronger in PRS than in ERA5 (Figs. 4b and
4d). Equally, the upper troposphere shifts again to cold core around 2000UTC on 08-Nov when
Rolf made a landfall partially over southern France (Fig. 3b).


Along this cyclone track, a time sequence of SLP of the cyclone centre is given in Fig. 5a. In

ERA5, the SLP drops down to 996 hPa at 03-06-Nov when the cyclone still has a deep cold core

(Fig. 4a). After this peak, the SLP keeps increasing to 1006 hPa until 0900UTC on 07-Nov and

again the SLP centre shrinks down to 1002 hPa at 0300UTC on 08-Nov when Rolf has a peak of

deep convection (e.g., Dafis et al., 2018) and a deep warm core is structured through the entire

troposphere (Figs. 4a and b). In the simulated Rolf, the SLP also drops down to 991 hPa at

0600UTC on 06-Nov during the preconditioning period of the tropical-like cyclone. Similar to

ERA5, the SLP of the cyclone increases to 996 hPa until 0000UTC on 07-Nov, decreases again, and

consequently the deepening of the low pressure reaches 990 hPa between 0000 and 0300UTC on

08-Nov (Fig. 5a). After this peak, the SLP of the simulated cyclone increases rapidly approaching

southern France (the depression is weakened to 1002 hPa after 20-08-Nov). The development of the

cyclone can be partially linked with the water vapour gained by the cyclone as shown in Fig. 5b. In

ERA5, the latent heat flux gradually increases from the beginning until 0000UTC on 07-Nov (from

100 to 220 $Wm^{-2}$). After 0600UTC on 07-Nov, the latent heat flux gradually decreases and slightly

increases until 03-08-Nov when the SLP depression is maximum. The latent heat flux decreases

gradually again following the approach to the partial landfall over southern France. In PRS as well,

the latent heat flux gained by the cyclone increases from the beginning until 06-07-Nov (from 140

to 320 $Wm^{-2}$) even though this is stronger than in ERA5. After this peak, the latent heat flux begins

to reduce and is slightly enhanced at 00-08-Nov. Similar to ERA5, the latent heat flux drops again

until the landfall over southern France. Surface flux and the corresponding diabatic heating is an

energy source of transition from extratropical to subtropical and tropical-like cyclones (e.g.,

Emanuel, 2005; Quitián-Hernández et al., 2020). Rolf also obtains a huge amount of water vapour

from the underlying sea surface during its phase transition and development. The difference in

intensity and transition timing between ERA5 and PRS may be caused by the difference in

evaporation and condensation the cyclone gains. However, PRS is able to realistically reproduce the

medicane Rolf and we will investigate the impact of climate change on Rolf in the next section.

**4. Simulation of Medicane Rolf under 1.5$^{o}$C global warming**

As explained in Section 2, we explore how Rolf is affected by the future climate change

(middle of the 21$^{st}$ century), which corresponds to global warming of 1.5$^{o}$C using the pseudo global

warming (PGW: e.g., Schär et al., 1996; Rasmussen et al., 2011; Parker et al., 2018) technique. In

addition to the effects of climate change, the relative roles of the atmosphere and the ocean in the

modulations of medicane Rolf are also investigated separately in this section.

Figure 6 shows the simulated cyclone tracks of Rolf in the PGW experiments. PGW$_{ALL}$

reproduces quite a similar cyclone track to that in PRS (Figs. 6a). From the beginning to 0000UTC





on 07-Nov, the cyclone moves southward approaching the Balearic Islands. After 0000UTC on 07-
Nov, the cyclone progresses northward and makes landfall at 2100UTC 08-Nov. While this
behaviour is not considerably different from that in PRS, a few other differences can be detected.
Under the future climate change, Rolf does not make landfall over Ibiza and the latitude where the
cyclone shifts its direction from south to north is relatively higher than that in PRS (Fig. 6a). The
cyclone track shifts more westward approaching southern France (4°E in $PGW_{ALL}$ and 6°E in PRS)
and the cyclone makes landfall slightly earlier than PRS. These modifications in the cyclone track
are more remarkable in $PGW_{SST}$ shown in Fig. 6b. The simulated medicane changes its marching
direction to the north at much higher latitude (higher than 40°N), far from the Balearic Islands, at
0000UTC on 07-Nov. After this shift, the cyclone moves northward similar to PRS and $PGW_{ALL}$,
but the timing of direction change is earlier by 6 hours and its direction shifts more westward than
PRS and $PGW_{ALL}$ (Rolf in $PGW_{ALL}$ begins to move westward after 0000UTC on 08-Nov in Fig.
6a). Due to those modifications, the simulated medicane achieves landfall over southern France at
3.8°E (slightly more western than $PGW_{ALL}$) and the landfall is much earlier than PRS and $PGW_{ALL}$,
which is at 1200UTC on 08-Nov. Interestingly the $PGW_{ATMS}$ simulation of Rolf shows a clearly
opposite tendency of changes to $PGW_{ALL}$ and $PGW_{SST}$ in Fig. 6c. The simulated cyclone strikes
Ibiza like PRS, but the cyclone in $PGW_{ATMS}$ progresses more southward while the cyclone in PRS
moves eastward after this landfall on Ibiza (Figs. 6c). The cyclone in $PGW_{ATMS}$ still moves
eastward after 0000UTC on 07-Nov and finally changes its direction to north at 1200UTC on 07-
Nov, which is later by 6 and 12 hours than PRS or $PGW_{ALL}$ and $PGW_{SST}$. Instead of moving
westward, the cyclone in $PGW_{ATMS}$ orientates to the northeast at 0000UTC on 08-Nov and
approaches south France around 7°E at 0000UTC on 09-Nov shown in Fig. 6c. The response of the
cyclone track to climate change seems different between $PGW_{ALL}/PGW_{SST}$ and $PGW_{ATMS}$ and we
see how the other features of the cyclone will change in the PGW experiments.

Figure 7a gives a time series of SLP in the cyclone centre of PGWs along the cyclone tracks

in Figs. 6. Rolf in $PGW_{ALL}$ develops the SLP centre in quite a similar way to Rolf in PRS. In the
beginning, the SLP depresses once to 991 hPa at 0700UTC on 06-Nov and increases the SLP of the
centre until 0000UTC on 07-Nov. Again, the SLP reduces and reaches the other minimum of
990hPa at 0000 to 0300UTC on 08-Nov. While the SLP of the cyclone centre is almost identical
between PRS and $PGW_{ALL}$, the strength of deepening is different. Figure S1 shows the scalar of
SLP gradient for PRS and $PGW_{ALL}$ at 0000UTC on 08-Nov. It is obvious that the SLP gradient is
much stronger in the $PGW_{ALL}$ than in the PRS around the peak time indicating that the warmer
climate tends to deepen the centre of the medicane, which could be linked to the changes in wind
and precipitation (described later). Such a modest enhancement of the medicane has been concluded
by Cavicchia et al. (2014) by climate projection experiments. Compared to PRS, the cyclone in


PGW$_{ALL}$ decays relatively rapidly after the peak at 0300UTC on 08-Nov, in particular, after
1200UTC on 08-Nov. This is likely to be due to the earlier time of landfall of the PGW$_{ALL}$ cyclone
over southern France (Figs. 3b and Fig.6a). Inversely, the SLP of the PGW$_{SST}$ cyclone drops down
intensively to 985 hPa from the beginning to 0700UTC on 06-Nov. The SLP centre in the PGW$_{SST}$
does not increase and continues to reduce the SLP until 1900UTC on 07-Nov to 963 hPa, which is
the earlier peak time and much stronger depression of the cyclone centre than those in PRS and
PGW$_{ALL}$. After this peak, the cyclone in the PGW$_{SST}$ decays quite rapidly (approximately 20 hPa
per 12 hours between 0000UTC and 1200UTC on 08-Nov) associated with the earlier landfall time
than PRS and PGW$_{ALL}$ (Figs. 3b, 6a and 6b). After the landfall, the cyclone continues to decay
further and almost disappears at 2300UTC on 08-Nov over southern France. It is of interest that in
PGW$_{ATMS}$ the depression of SLP is substantially reduced throughout cyclone tracking. While In the
beginning, the SLP of the cyclone centre is almost identical with those of PRS and PGW$_{ALL}$, the re-
depressing of the cyclone centre after 0000UTC on 07-Nov is much weaker than PRS and PGW$_{ALL}$.
The second peak of low SLP is detected at 2300UTC on 07-Nov (slightly earlier than PRS and
PGW$_{ALL}$), but it shrinks only to 1000 hPa. The cyclone begins to decay gradually after the second
peak. Interestingly, this result suggests that the role of future climate change in the atmosphere and
ocean have competing effects on the medicane development.

Figure 7b gives a time series of latent heat flux gained (averaged) by the simulated cyclone

within a radius of 250 km. The evaporation in PGW$_{ALL}$ is relatively larger than that in PRS, in
particular, from the beginning to 1200UTC on 07-Nov (approximately 50 W/m$^2$ higher at largest).
The temporal variation in evaporation along the cyclone track is almost identical between PRS and
PGW$_{ALL}$. Correspondingly to the more rapid decay of the cyclone in PGW$_{ALL}$, the latent heat flux
in PGW$_{ALL}$ decreases more rapidly than PRS after 0000UTC on 08-Nov. In PGW$_{SST}$, the simulated
cyclone obtains much more water vapour from the underlying warmer SST. From the beginning,
the latent heat flux is about double that in PRS and increases up to 500 W/m$^2$ until 1200UTC on 07-
Nov continuing to 0000UTC on 08-Nov with the same amplitude. The uptake of water vapour
drops suddenly after 0000UTC on 08-Nov and becomes lesser than that in PRS at 0600UTC on 08-
Nov and is diminished to almost zero after the earlier landfall. Inversely, the evaporation in
PGW$_{ATMS}$ is inactivated compared to that in PRS during the entire period of cyclone tracking (100
W/m$^2$ at smallest). However, the temporal variation in evaporation is quite similar to that in PRS
having a peak around 06-07-Nov. The decreasing rate of the evaporation after the peak in PGW$_{ATMS}$
is relatively more moderate than those in PGW$_{ALL}$ and PGW$_{SST}$ due to the later time of the landfall
(Fig. 6c). While the uptake of water vapour differs among PGWs, its peak leads the maximum of
the medicane similarly by 6 to 12 hours (Figs. 7a and 7b).
In PRS, the precipitation associated with the cyclone is intense at 0000UTC on 06-Nov and
decreases until 1200UTC on 06-Nov in Fig. 7c. That could be associated with deep cumulus
convection due to the initial cut-off low and trigger of warm seclusion (e.g., Mazza et al., 2017; Fita
and Flaounas, 2018). After 1200UTC on 06-Nov, the precipitation remains in a relatively small
with some fluctuations before increasing again at 1200UTC on 07-Nov reaching a peak around
2100UTC on 07-Nov, which is somewhat earlier than the peak of SLP depression (Fig. 7a).
Coinciding with the reduction in the SLP depression, the precipitation decreases again after the
peak. The precipitation in PGW$_{ALL}$ shows quite a similar variation to that in PRS until 1200UTC on
07-Nov although the precipitation is slightly stronger. While the precipitation in PGW$_{ALL}$ is
reactivated at the same time as PRS, its amplitude of the peak around 21-07-Nov is much larger
than that in PRS. That is, the simulated cyclone in PGW$_{ALL}$ can obtain more energy from diabatic
heating than PRS resulting in a stronger deepening of SLP shown in Fig. S1. This stronger
precipitation can be associated with an enhanced uptake of the water vapour in PGW$_{ALL}$ as shown in
Fig. 7b. In PGW$_{SST}$, the precipitation until 1000UTC on 06-Nov varies quite similarly to that in
PRS and PGW$_{ALL}$, however the precipitation keeps its relatively strong intensity and consequently,
the difference from PGW$_{ALL}$ and PRS is large during the cyclone track. After 0000UTC on 07-Nov,
precipitation gets more activated and its peak reaches 2.7 mm/hour before 0000UTC on 08-Nov.
Similar to PGW$_{ALL}$, after the peak the precipitation is abruptly reduced due to the earlier timing of
the landfall (Fig. 6b). This intense rainfall can be associated with the fact that the cyclone is fueled
with abundant water vapour in PGW$_{SST}$ (Fig. 7b). The precipitation of PGW$_{ATMS}$ also shows an
identical variation with PRS in the beginning of the track. Associated with the moderate latent heat
flux in Fig. 7b, the precipitation is less during the whole lifecycle of the cyclone and has a peak at
1700UTC on 07-Nov with a smaller amplitude than those in PRS and other PGWs.
Figure 7d illustrates a time series of hourly maximum wind speed (MWS, hereafter) around
the cyclone in each simulation. The MWS is defined as a value of 10m-wind speed at a grid where
the maximum value is detected every hour within 250km radius of the cyclone. In PRS, from
0000UTC on 06-Nov until 0000UTC on 07-Nov, the MWS decreases and increases until 0800UTC
on 08-Nov. This variation is roughly consistent with that in the SLP (Fig. 7a). However, the MWS
has another peak at 1800UTC on 08-Nov. In PGW$_{ALL}$, the hourly changes in MSW are similar to
those in PRS, but that is stronger than in PRS through the most lifecycle (at largest, 6m/s higher in
PGW$_{ALL}$). After 1200UTC on 08-Nov, the MWS in PGW$_{ALL}$ abruptly dropped. This could be
caused by the earlier landfall in PGW$_{ALL}$ than PRS (Fig. 6a) and therefore, the second peak of the
MWS might be missed in PGW$_{ALL}$. In PGW$_{SST}$, the MWS does not decrease, but that is reinforced
from the beginning until 2200UTC on 07-Nov. While the MWS exceeds to 40 m/s at 1600UTC on
07-Nov and commences to decrease gradually at 0000UTC until 1500UTC on 08-Nov, the MWS



falls down rapidly afterward due to the earlier landfall (Fig. 6b). In PGW$_{ATMS}$, during 06 and 07-
Nov, the MWS is slightly stronger than that in PRS (but, weaker than PGW$_{ALL}$). However, the grid
number of high wind speed is much less in PGW$_{ATMS}$ than in PRS (not shown here, but a plot of
horizontal structure of surface wind speed will be given in Fig. 9). After 0000UTC on 08-Nov (the
deep warm core is well difined), the MWS in PGW$_{ATMS}$ is weaker than that in PRS.

Figure 8 illustrates a diagram of the cyclone phase space in PGWs. Whilst the phase shift

from a shallow to a deep warm core is almost identical in the PRS and PGW$_{ALL}$, the warm core of
PGW$_{ALL}$ simulated cyclone is relatively stronger around the peak of SLP depression (03-08-Nov in
Fig. 7a), particularly, in the lower troposphere. This stronger warm core is consistent with the
enhanced deepening of the cyclone shown in Fig. S1 and with the enhanced precipitation in Fig. 7c.
From 1500UTC to 2100UTC on 08-Nov, the structure of warm core is diminished in PGW$_{ALL}$ and
this is due to the earlier landfall than PRS. The cyclone in PGW$_{SST}$ changes its phase from shallow
to deep warm core at 0600UTC on 06-Nov (Fig. 8b), which is much earlier than PRS and PGW$_{ALL}$
(at 1500UTC to 1800UTC on 06-Nov in Figs. 4b and 7a). Once the cyclone shifts to tropical-like
features, the structure of the deep warm core is strengthened very rapidly and consequently, the
simulated cyclone is matured with a much larger value of phase space than PRS and PGW$_{ALL}$. In
turn, the tropical-like structure shrinks abruptly after its mature state and eventually the cyclone is
reduced to one with a cold core at 2100UTC on 08-Nov corresponding to the earlier landfall. The
phase shift of the cyclone in PGW$_{ATMS}$ is similar to those in PRS and PGW$_{ALL}$ in Fig. 8c. In
contrast, after the cyclone is converted into a tropical-like cyclone from 1200UTC to 1500UTC on
06-Nov, the development of a deep warm core stagnates in the period of November 7th. The
enhancement of deep warm core can be found after the stagnation. However, the cyclone gets its
deep warm core matured in a moderate value of the space. There is no rapid reduction of warm core
in PGW$_{ATMS}$ since the cyclone achieves the landfall later than other PGWs Fig. 6c.

Under global warming, the development of the medicane is modified with respect to that at

present (in particular, a moderate intensification as aforementioned from Figs. 6 to 8). Here, we
explore the horizontal structure of the medicane. The wind speed of PRS exceeds to 24 m/s at the
peak of SLP depression (based on Fig. 7a) in Fig. 9a. The area of high wind speed spreads widely
(more than 100 km radius of the cyclone). In PGW$_{ALL}$, while the radius of high wind speeds
appears to be slightly small, the wind speed is 24 m/s over a large part within the radius of 100km
(Fig. 9b) and the maximum values (faster than 26 m/s) is larger than that of PRS. This result is
consistent with the stronger deepening of the cyclone centre in PGW$_{ALL}$ as shown in Fig. S1.
Regarding the extreme intensification in the SLP depression, the surface wind speed is much
stronger in PGW$_{SST}$ than PRS and PGW$_{ALL}$ in Fig. 9c. The wind speed exceeds to 30 m/h
everywhere within the radius of 100km (except for the centre) and the area where the wind speed is





larger than 20 m/s extends to the radius of 150km. In contrast, wind speeds for the cyclone in
$PGW_{ATMS}$ are substantially lower. Its maximum of wind speed is 24 m/s, which is equivalent to that
in PRS (as shown Fig. 7d, the hourly MWS in $PGW_{ATMS}$ is slightly larger than that in PRS), but the
area of high speed winds is obviously diminished in Fig. 9d and the strong wind speed is limited
only in the northern sector around the centre (similar spatial limitation can be seen in hourly MSW
in $PGW_{ATMS}$, not shown). Compared to PRS and other PGWs, the size of the simulated cyclone
tends to become smaller in $PGW_{ATMS}$. These changes in intensity of the cyclone correspond well
with those of the cyclone phase space (Fig. 8). It is interesting to note the hurricane-like structure
and intensity of the cyclone in $PGW_{SST}$.

Figure 10 illustrates the rainband structure of each simulated cyclone during the

precipitation peak given in Fig. 7c. In PRS, the cyclone has a spiral band of precipitation around the
centre (Fig. 10a). In particular, the precipitation is active (up to 12 mm/h) in the northern sector of
the cyclone and the strong rainfall extends to the northeast direction. There is little rainfall in the
centre area, which is cloud-free "eye"; this can be easily detected and it is also a key feature of
tropical-like cyclones. As seen in Fig. 7c the precipitation of $PGW_{ALL}$ intensifies during its peak in
Fig. 10b. Whereas the spiral band of precipitation is likely to be similar to that in PRS in the
northern sector, the northeastward-orientated rainband is enhanced significantly (more than 18
mm/h). In addition, the precipitation is also more vigorous than PRS in the southern sector, in
particular, around the centre the precipitation is more than 10 mm/h and the spiral rainband of the
medicane is also reinforced due to projected global warming. The eye of the medicane is identical
to that in PRS. The warmer SST enhances the spiral band more effectively in Fig. 10c as shown in
Fig. 7c. The precipitation around the centre exceeds 20 mm/h in the southern sector and the
northeastward rainband is elongated with intense rainfall. In the far side of the southern sector, the
rainband is more activated compared to $PGW_{ALL}$. Interestingly, the eye of the medicane becomes
larger than that in PRS and $PGW_{ALL}$ and is more clearly organized. This is associated with the
much deeper depression of SLP in $PGW_{SST}$ (Fig. 7a). Corresponding to the deactivated precipitation
due only to the warmer atmosphere (Fig. 7c), the rainband around the cyclone centre in $PGW_{ATMS}$ is
reduced significantly as shown in Fig. 10d. While the maximum rainfall is still more than 10 mm/h
near the centre, the rainband almost loses its spiral structure and the area of vigorous rainfall
decreases. It seems that the eye of the cyclone still survives, but that forms less clearly than other
cases. As shown in Fig. S2, the medicane in $PGW_{SST}$ has much a larger cloud-free eye around the
centre than PRS and other PGW experiments (witnessed by outgoing longwave radiation) and this
intensified medicane can be classified into a hurricane.

**5. Discussion on relative role of warmer atmosphere and ocean in medicane development**



In the previous section we showed that the warmer climate leads to a moderate medicane
intensification in agreement with previous studies (e.g., Cavicchia et al., 2014; Tous et al., 2016;
González-Alemán et al., 2019). The results also showed more enhanced precipitation, surface wind
speed and a SLP deepening around the medicane. Interestingly though, the warmer atmosphere
inhibits the medicane development substantially, while the warmer ocean reinforces the medicane
dramatically. In this section, we discuss the factors that underlie the different roles of the
atmosphere and the ocean in the response of the medicane development to future warming.
Figure 11a gives a time function of convective available potential energy (CAPE) averaged
within the 250km radius around the cyclone centre. CAPE in PRS increases from the beginning and
reaches its peak around 1000UTC on 07-Nov. This peak occurs earlier than the maximum of
precipitation as shown in Fig. 7c. In the remaining time, CAPE decreases corresponding to the
decay of the cyclone. PGW$_{ALL}$ has a slightly larger CAPE than PRS in particular before the peak of
SLP depression. Because the cyclone in PGW$_{ALL}$ makes an earlier landfall, CAPE also drops more
rapidly than PRS. Such a difference is most obvious in PGW$_{SST}$. CAPE in PGW$_{SST}$ becomes much
larger at 0600UTC on 06-Nov and the timing of its peaks is relatively earlier than PRS. After the
peak, CAPE decreases much more abruptly than PGW$_{ALL}$ due to the earliest time of landfall.
Inversely, increase ratio of CAPE in PGW$_{ATMS}$ is more moderate and decaying of CAPE is also
more slowly than PRS. Figures 11b-e give CAPE of each WRF simulation at its maximum in Fig.
11a. Between PRS and PGW$_{ALL}$, the cyclone gains more energy in PGW$_{ALL}$ (Figs. 11b and 11c)
resulting in the enhanced precipitation and consequently stronger SLP deepening (Fig. 7). In
PGW$_{SST}$, the medicane is also fueled much energy like PGW$_{ALL}$ and the area of large CAPE spreads
more widely around the cyclone centre than PRS and PGW$_{ALL}$ (Fig. 11d). This wider area of high
CAPE can be consistent with the larger area of high wind speed (Fig. 9c). Contrastingly, CAPE in
PGW$_{ATMS}$ shrinks extensively and its size of high CAPE is much smaller than PRS.
CAPE is a physical indicator that serves to estimate to what extent energy can be utilized for
cumulus convection. In PGW$_{SST}$, the cyclone is fueled by increased water vapour mainly due to
underlying warmer SST (Figs. 2b and 7b). As such the air mass can be saturated more easily than in
PRS when background tropospheric temperature and humidity is identical in the two simulations
(here, the temperature/humidity of lateral boundary condition is regarded as background
temperature/relative humidity, see Section 2). This situation is favourable to gain more CAPE and
therefore, PGW$_{SST}$ has much higher CAPE than PRS (Figs. 11a, 11b and 11d). Since the diabatic
heating is a source of energy for cyclone development, more CAPE and precipitation enhance the
cyclone and the surface wind also increases. This stronger surface wind, in turn, activates more
evaporation from the sea surface (via WISHE feedback) and, consequently, the cyclone can be
moistened more effectively. It is noticeable that the medicane in PGW$_{SST}$ consumes CAPE more
rapidly than in other PGW experiments, which indicates that the WISHE mechanism works more
effectively (also evidenced by much stronger MWS in $PGW_{SST}$). Consequently, the medicane in
$PGW_{SST}$ can have a hurricane-like structure of surface wind speed, rainfall, and OLR (Figs. 9c, 10c,
and S2c). Conversely, in $PGW_{ATMS}$, the background troposphere is warmed and drier through the
entire troposphere compared to PRS (Fig. 2c). Even though the ocean forcing is similar in PRS and
$PGW_{ATMS}$ (since the SST boundary condition does not differ), the warmer temperature and the
lower relative humidity due to global warming (Fig. 2c) is unfavorable for condensation. That is,
the warmer and drier atmosphere can inhibit cumulus convection and CAPE is reduced. As a result,
the diabatic heating is less effectively generated and the SLP depression and corresponding WISHE
feedback are also deactivated in $PGW_{ATMS}$ (the hourly MWS is slightly higher in $PGW_{ATMS}$ than
PRS before the peak, but the largest MWS is weaker in $PGW_{ATMS}$ than PRS and the area of high
wind speed is very limited in $PGW_{ATMS}$ compared to PRS shown in Figs. 7d, 9a, and 9d). The
moderate intensification of the medicane in $PGW_{ALL}$ is a consequent of the competition between
enhancement due to the warmer SST and suppression due to the warmer/drier atmosphere.
However, we need to consider a role of SST change due to surface wind and evaporation. When
evaporation is more effective in $PGW_{SST}$ and less in $PGW_{ATMS}$, the underlying SST can be cooled
down and warmed up. That is, the results shown in this study do not contain all the process of air-
sea interaction for the impacts of a warmer climate on the medicane. Therefore, we will need to
investigate future changes of the medicane with an atmosphere-ocean coupled model (e.g., Akhtar
et al., 2014; Mooney et al., 2016; Ricchi et al., 2019) in the future to increase robustness of our
results in this study.

**6. Concluding Remarks**

In this study we investigated the impacts of future climate change on a tropical-like cyclone

(medicane) formed in the Mediterranean Sea in a PGW framework with the WRF regional climate
model. The main novelty of this work is the investigation of the relative roles of the atmosphere and
ocean, respectively in the medicane's response to projected global warming.

Based on the assessment experiments for better combination of three physical

parameterizations of WRF (Tiedtke for cumulus convection, Thompson for microphysics, and
Mellor-Yamada-Nakanishi-Niino for planetary boundary), we simulated the medicane Rolf under
present (PRS) and middle future climate adapting PGW technique (e.g., Parker et al., 2018; Mooney
et al., in review). In ERA5, the cyclone gradually shifts its phase from extratropical cyclone (deep
cold core) to a shallow warm-core cyclone. This finally transitions into a deep warm-core cyclone
around 0000UTC on 8[th] November. After this peak, the cyclone becomes weaker (the SLP
depression is reduced). Compared to the best track of ERA5 reanalysis, PRS of WRF simulates Rolf




realistically making a landfall over southern France. While the intensity of Rolf is stronger in PRS
than in ERA5 partially because of difference in grid size, the SLP deepening decreases to 990 hPa
in PRS, which is consistent well with previous studies (e.g., Miglietta et al., 2013). PRS also
represents well the phase transition to a tropical-like cyclone even though its conversion into the
tropical-like is achieved earlier in time than ERA5.
The PGW experiments revealed obvious changes in medicane structure associated with
global warming. First, there is a clear impacts on the cyclone track: in PGW$_{ALL}$ and PGW$_{SST}$, the
medicane tends to march over more northern and western pathway and its timing of landfall
becomes earlier than PRS. Conversely, the medicane in PGW$_{ATMS}$ shifts more southward and
eastward. This difference in cyclone track might not be a random response, but seems to be
associated with changes in the intensity of the medicane. In PGW$_{ALL}$ and PGW$_{SST}$, the medicane is
more enhanced in terms of surface wind and precipitation around the cyclone centre (e.g., Cavicchia
et al., 2014; González-Alemán et al., 2019) and the degree of intensification is much stronger in
PGW$_{SST}$ and PGW$_{ALL}$ (e.g., the hourly maximum wind speed exceeds to 40 m/s in PGW$_{SST}$ in Fig.
7d). The cyclone track of the stronger medicane in PGW$_{SST}$ is more to the north and, consequently,
makes an earlier landfall than in PGW$_{ALL}$. Inversely, the medicane in PGW$_{ATMS}$ reduces its
intensity to a large extent and perhaps this simulated cyclone can be categorised into a lower
category of medicane (i.e. tropical depression), for example, an unclear eye forming in the centre
and the smaller size of region with high wind speed. The northward shift in position of the
maximum wind speed associated with the medicane is also detected in a climate projection by Tous
et al. (2016). The changes in cyclone track shown in this study might be indicative for the results of
Tous et al. (2016). However, since our simulations address only one medicane, we will need to
investigate the changes in cyclone track due to global warming in other study cases, so that the
implication becomes more robust.
Our PGW simulations elucidated the counteracting individual contributions of a warmer
atmosphere and a warmer ocean to the development of medicane associated with the global
warming. Since the warmer and drier atmosphere inhibits cumulus convection indicated by weaker
CAPE, the energy due to diabatic heating is not sufficient. This situation can be ineffective to drive
the wind-induced surface heat exchange (hourly maximum wind speed is approximately equivalent
between PRS and PGW$_{ATMS}$, but the area of high wind speed is much smaller in PGW$_{ATMS}$ than in
PRS). Consequently, the transition from a cut-off low into a tropical-like cyclone tends to be
degraded; this is supported by our analysis of PGW$_{ATMS}$. Conversely, the warmer ocean surface
enriches the medicane with moisture, which allows cumulus convection to develop more
effectively. With a more efficient energy gain, the medicane growth is enhanced and WISHE (e.g.,
Eamanuel, 1986) can be also activated, as indicated by the results of PGW$_{SST}$. Consolidating these



reversal effects of warmer (and drier) atmosphere and ocean (through nonlinear processes), the
medicane intensifies to a moderate extent by global warming. While the medicane under global
warming shows a modest intensification in terms of wind speed and SLP deepening, precipitation
presents radical changes during the peak of intensity. This suggests that medicane could be more
hazardous due to global warming as concluded by González-Alemán et al. (2019).
In this study we have presented novel findings regarding the relative role of atmosphere and
ocean in the modulation of medicane development under global warming. It would be interesting to
see if other cases of medicanes show a similar response to the warmer atmosphere and ocean. For a
better quantification of changes, the simulation and investigation with a regional coupled model for
several cases will be desired in the future.

**Acknowledgement**
This study has been carried out under IBERTROPIC project (grant agreement no. CGL2017-
89583-R), funded by the Spanish Ministry of Science, Innovation and Universities, the Spanish
State Research Agency and the European Regional Development Fund. Koseki S. is supported by
Giner de los Ríos 2018/2019 and 2019/2020, which is a scholarship grant by la Universidad de
Alcalá. González-Alemán J.J. has been funded through grants BES_2014-067905 and FJC2018-
035821-I by the Spanish State Research Agency. The computational resource comes from the
Norwegian High-Performance Computing Program resources (NS9039K).

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





**Figures**

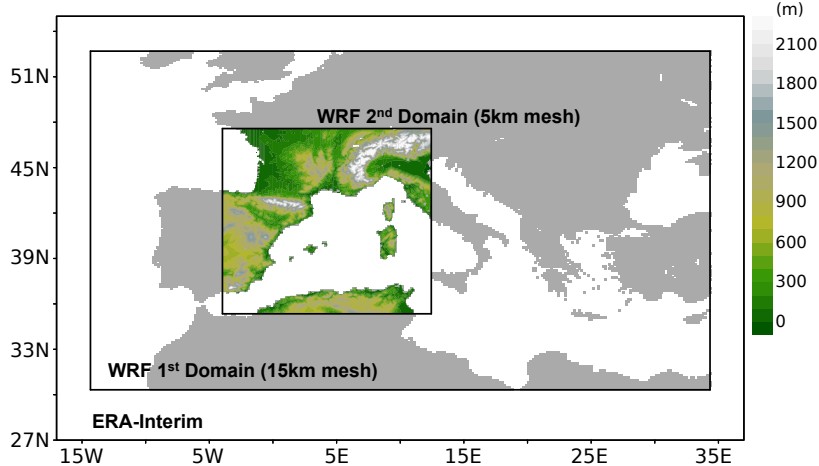

**Figure 1.**
Domains for WRF simulations for medicane, Rolf. Shading in WRF 2nd domain is topography height
from MODIS.


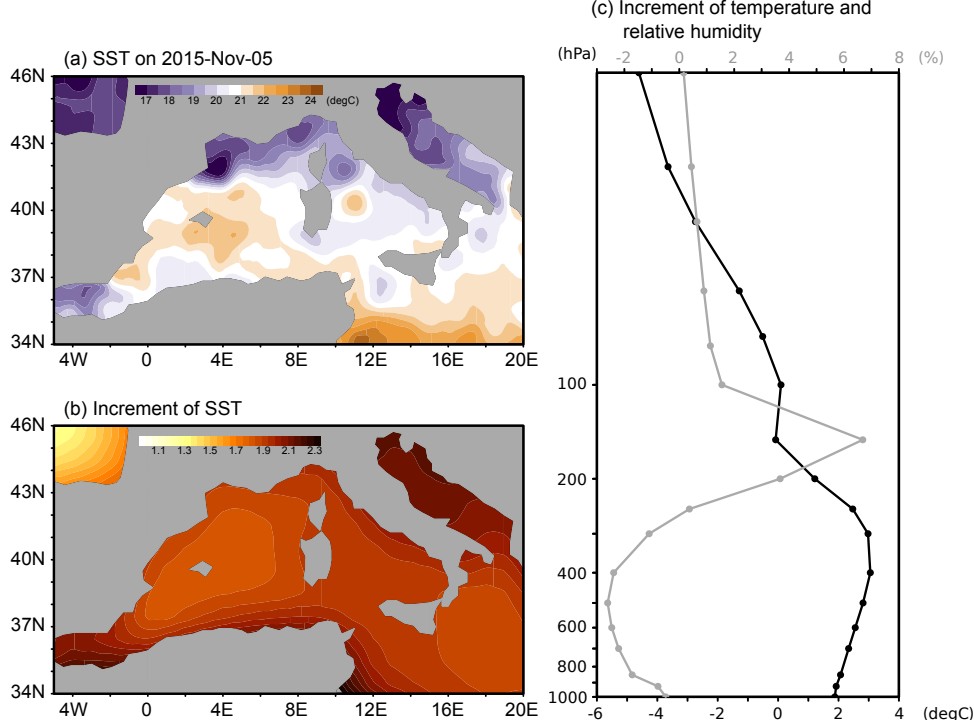

**Figure 2.**
(a) Sea surface temperature (SST) at 00UTC on 5[th] November, 2011 in OISST. Increment projected
by 18 CMIP5 CGCMs (b) SST and (c) vertical profiles of increment of air temperature
and relative humidity averaged over WRF's 2[nd] domain between 2035-2065 and 1975-2006.



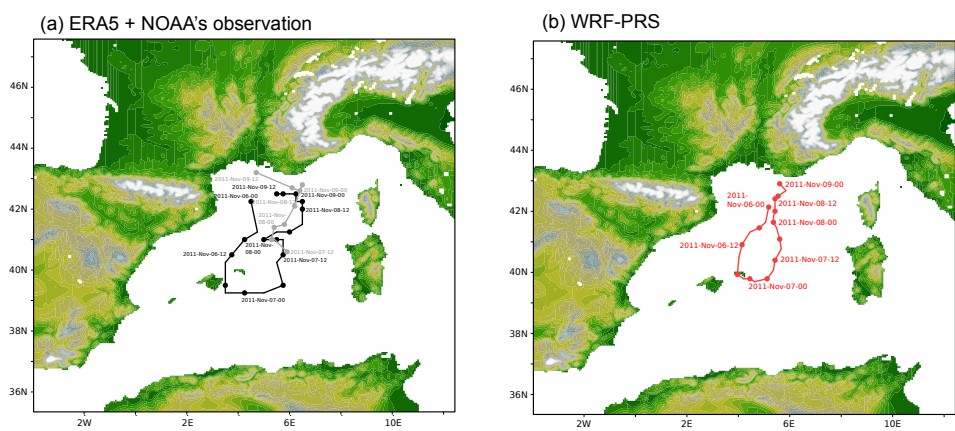

**Figure 3.**
Trajectory of medicane Rolf by (a) ERA5 (black) and NOAA observation (gray), and (b) PRS,
from 00UTC, 6th, Nov, 2011 to 12UTC, 9th, Nov 2011 (for NOAA observation, the trajectory is from
12UTC, 7th, Nov, 2011 to 12UTC, 9th, November, 2011). The tracking is based on the lowest sea level pressure.
Note that the track of PRS is until 00UTC, 9th, Nov 2011 because of early landfall on South France.



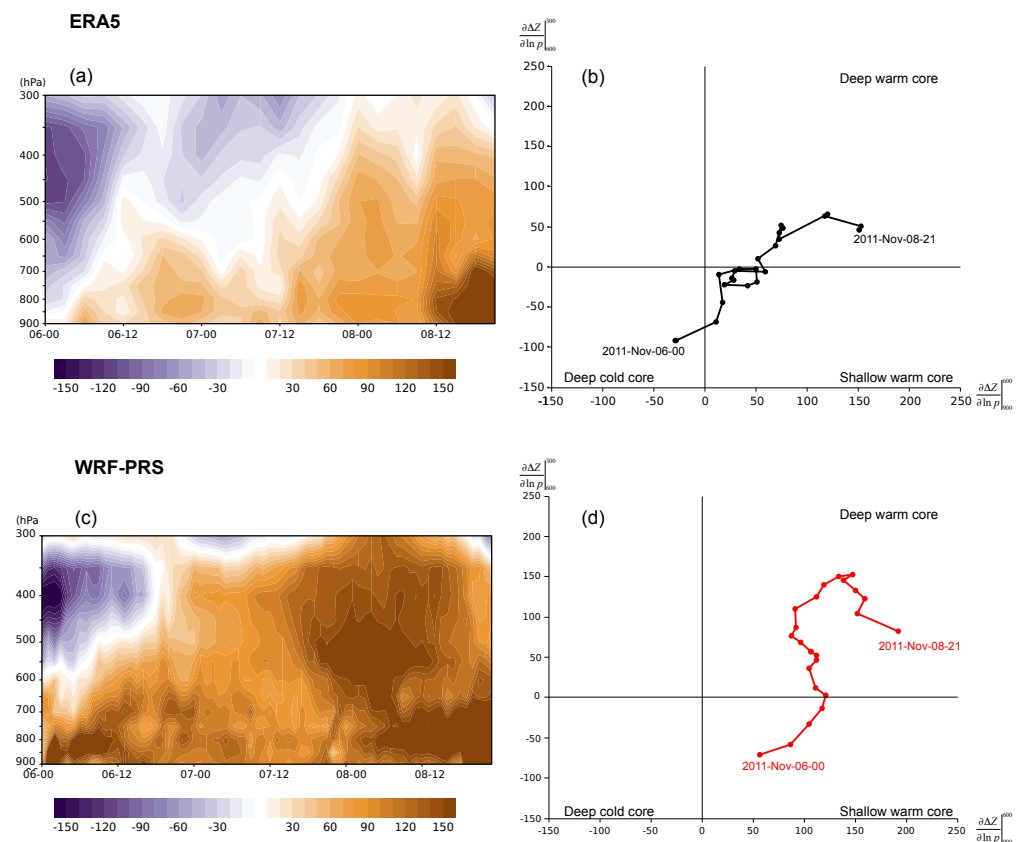

**Figure 4.**
Cyclone phase space defined by Hart 2003. (a) The pressure-time section of cyclone phase space for ERA5.
The index is estimated every 50 hPa. (b) The index is projected on upper (600-300 hPa) and lower (900-600 hPa)
phase in ERA5. (c) and (d) Same as (a) and (b) except for the WRF PRS simulation.




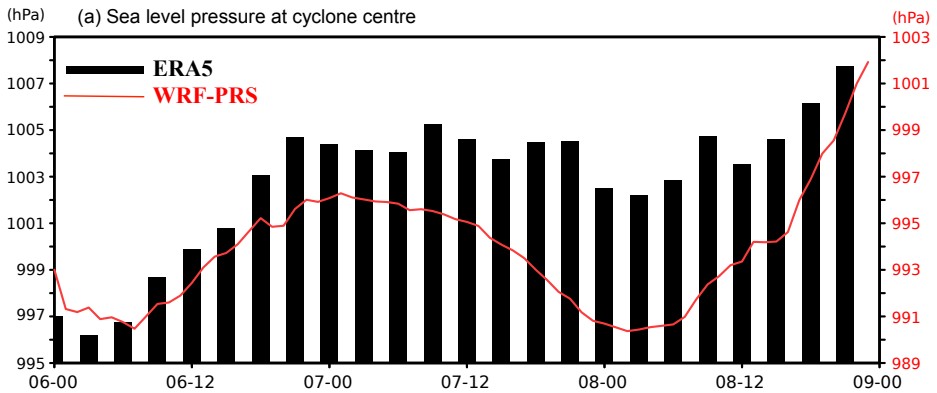

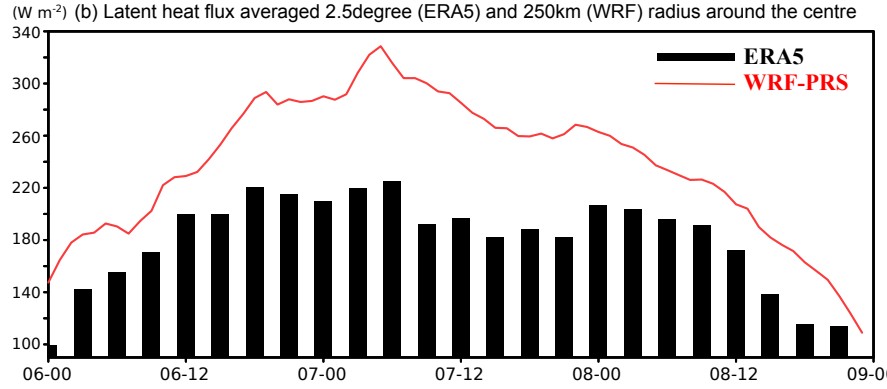

**Figure 5.**
Time series of (a) sea level pressure (SLP) at grid of cyclone centre and (b) latent heat flux
averaged within a radius of 2.5 degrees (ERA5) and 250km (WRF-PRS). The black bar and red line
denote the variables of ERA5 and WRF-PRS, respectively. Note that the labels in Fig. 5a for ERA5 and
PRS are on left and right hand side, respectively.



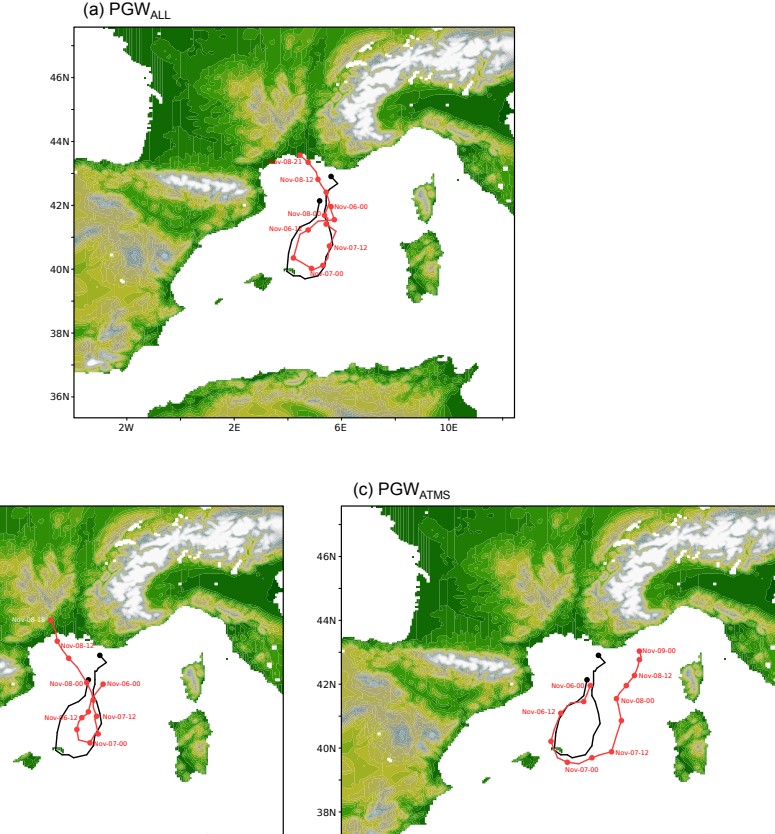

**Figure 6.**
Same as Fig. 3, but for (a) PGW$_{ALL}$, (b) PGW$_{SST}$, and (c) PGW$_{ATMS}$, respectively.
The black line is the cyclone track for PRS.



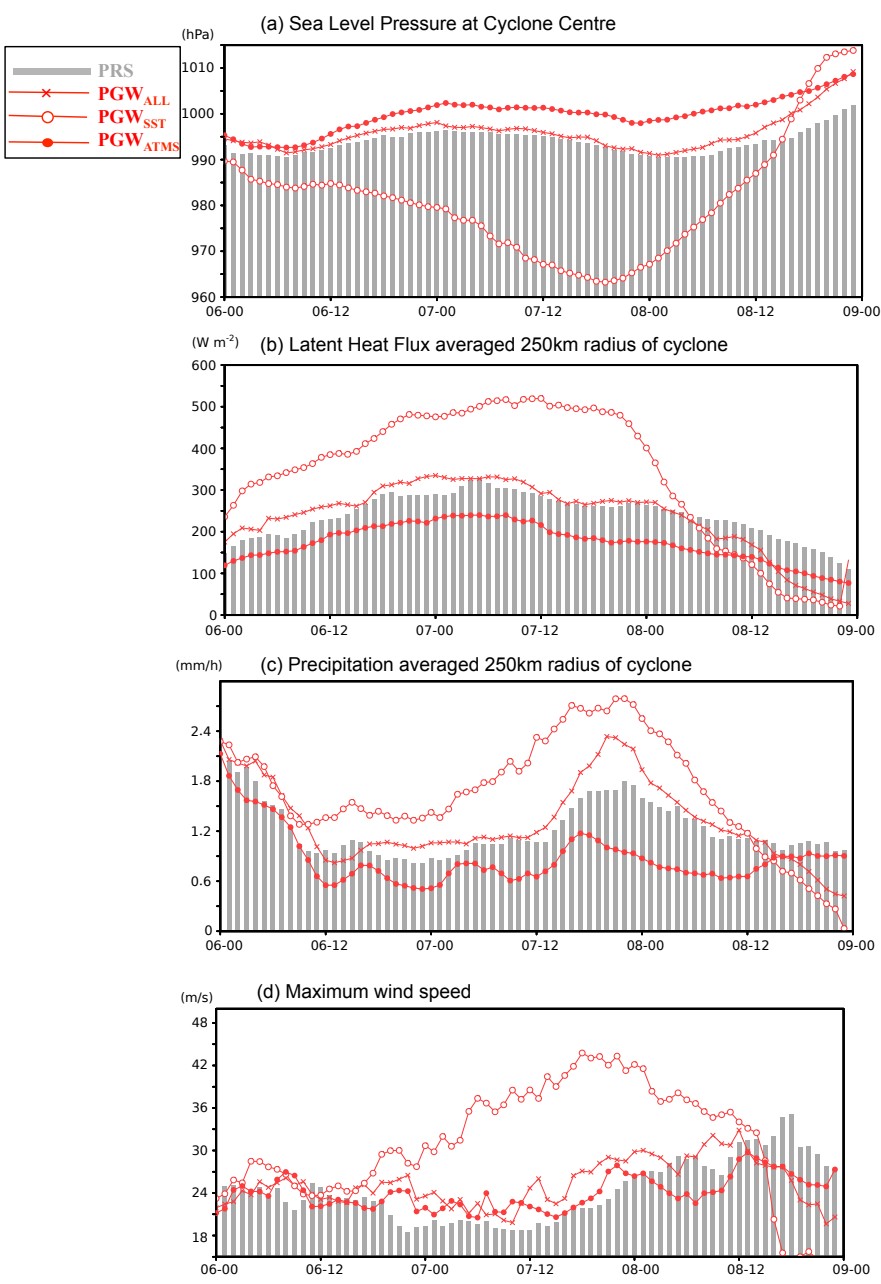

**Figure 7.**
Time series of (a) SLP at grid of cyclone centre, (b) latent heat flux, (c) precipitation averaged, and (d) maximum wind speed within the 250km radius of the simulated medicane. (b) and (c) are averaged value within 250k radius The gray bar and red lines denote the variables of PRS and PGWs (with different markers), respectively.


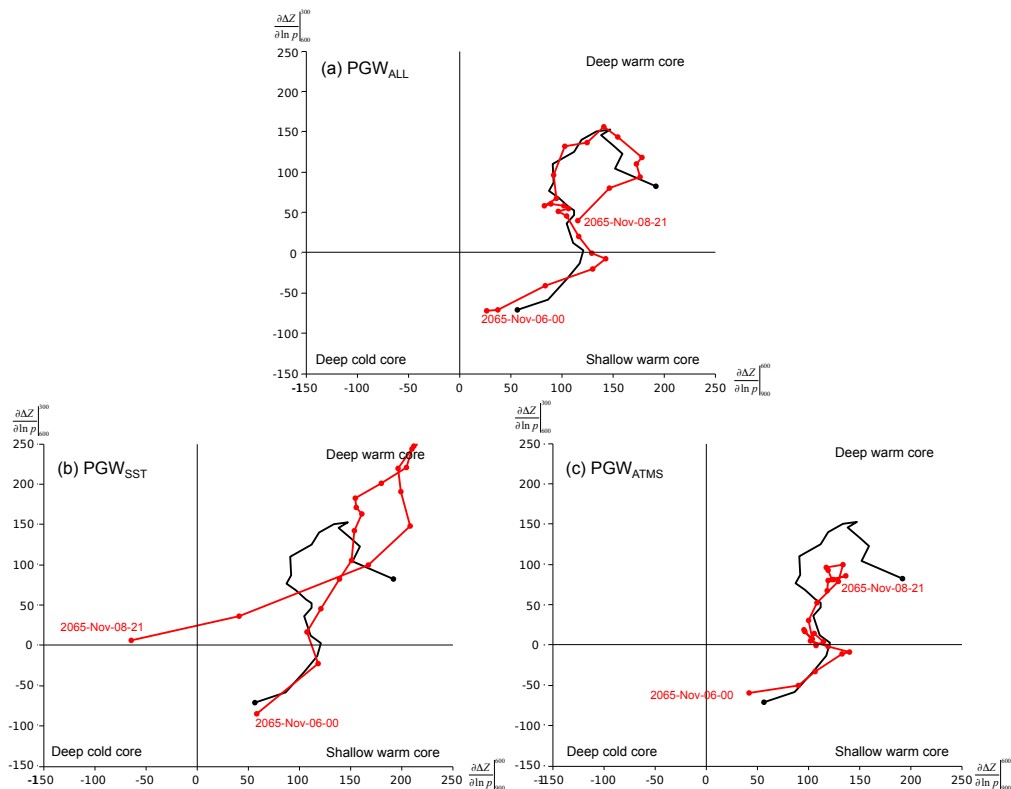

**Figure 8.**
Same as Fig. 4d, but for (a) PGW$_{ALL}$, (b) PGW$_{SST}$, and (c) PGW$_{ATMS}$, respectively. The index of PRS
is superimposed by black line.

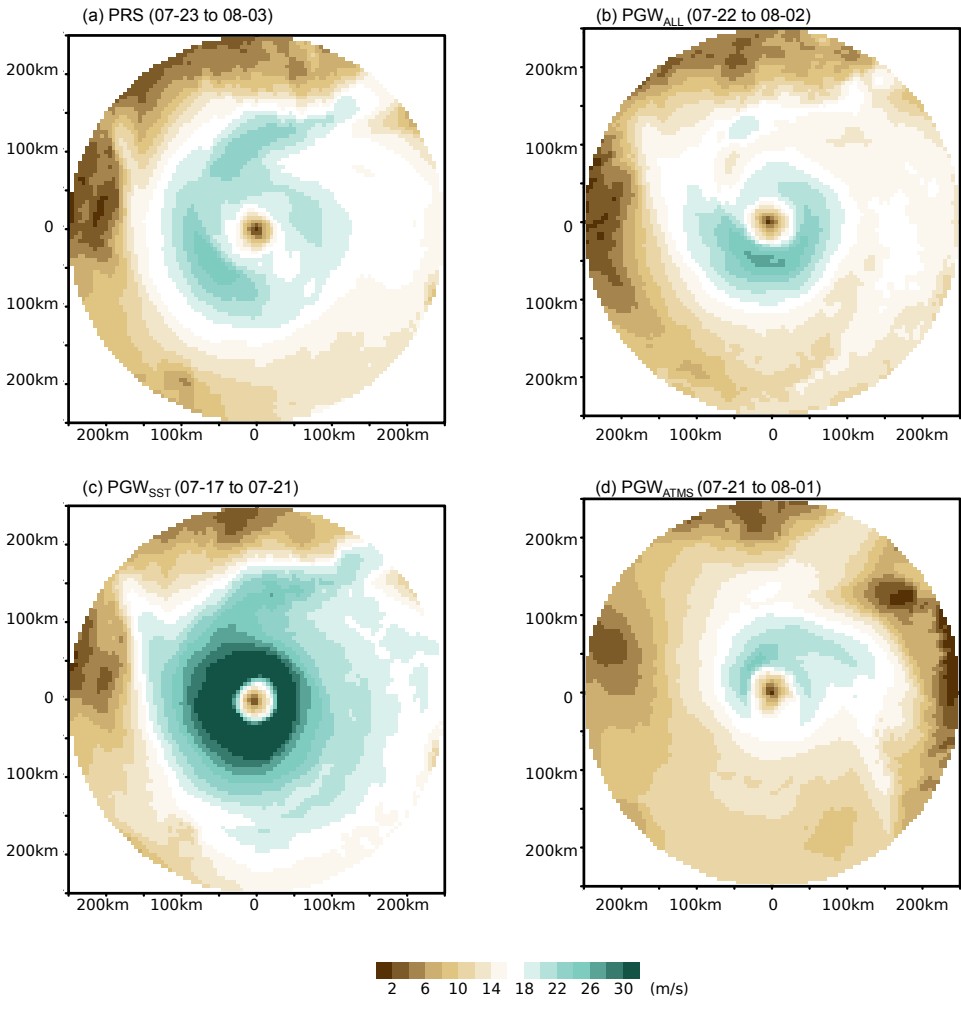

**Figure 9.**
Surface wind speed during SLP minimum (referring to Fig. 7a) for (a) PRS, (b) PGW$_{ALL}$, (c) PGW$_{SST}$, and (d) PGW$_{ATMS}$ around the cyclone centre, respectively.


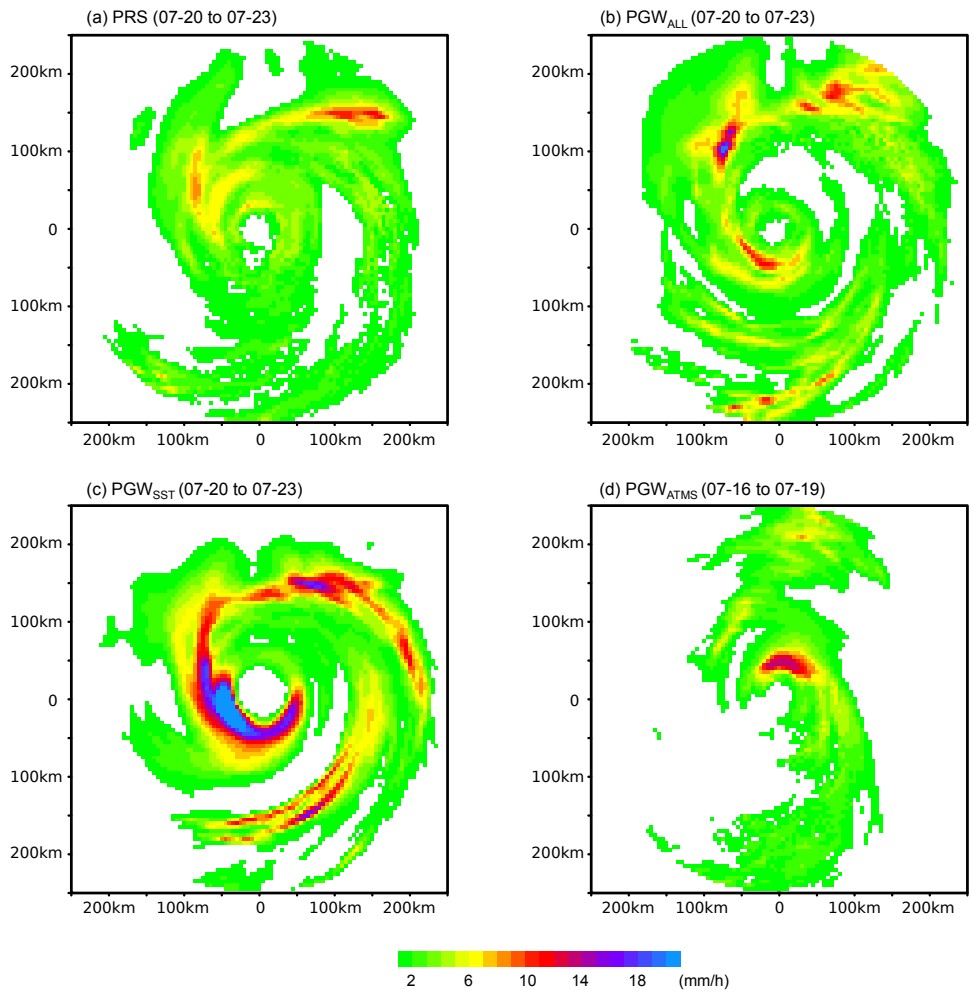

**Figure 10.**
Precipitation during its minimum (referring to Fig. 7c) for (a) PRS, (b) PGW$_{ALL}$, (c) PGW$_{SST}$, and (d) PGW$_{ATMS}$ around the cyclone centre, respectively.



**Figure 11.**
(a) Same as Fig. 7b, but for convective available potential energy (CAPE) and CAPE at its minimum
(referring to Fig. 7c) for (b) PRS, (c) PGW$_{ALL}$, (d) PGW$_{SST}$, and (e) PGW$_{ATMS}$ around the cyclone centre,
respectively.








**Table**

| Model Name | No. Ensemble Members from Historical Simulation | No. Ensemble Members from RCP8.5 Simulation | Ensemble Members Used | Names of Member Realisations |
|---|---|---|---|---|
| ACCESS1-3 | 3 | 1 | 1 | r1i1p1 |
| CanESM2 | 5 | 5 | 3 | r1i1p1, r2i1p1, r3i1p1 |
| CCSM4 | 6 | 6 | 3 | r1i1p1, r2i1p1, r6i1p1 |
| CESM1-CAM5 | 3 | 3 | 3 | r1i1p1, r2i1p1, r3i1p1 |
| CMCC-CM | 1 | 1 | 1 | r1i1p1 |
| CNRM-CM5 | 10 | 5 | 3 | r2i1p1, r4i1p1, r6i1p1 |
| CSIRO-Mk3-6-0 | 10 | 10 | 3 | r1i1p1, r2i1p1, r3i1p1 |
| GFDL-CM3 | 5 | 1 | 1 | r1i1p1 |
| GFDL-ESM2M | 1 | 1 | 1 | r1i1p1 |
| GISS-E2-H | 5 | 2 | 2 | r1i1p1, r2i1p1 |
| HadGEM2-CC | 3 | 3 | 3 | r1i1p1, r2i1p1, r3i1p1 |
| HadGEM2-ES | 4 | 4 | 1 | r3i1p1 |






| INM-CM4 | 1 | 1 | 1 | r1i1p1 |
|---|---|---|---|---|
| IPSL-CM5A-MR | 3 | 1 | 1 | r1i1p1 |
| MIROC5 | 4 | 3 | 3 | r1i1p1, r2i1p1, r3i1p1 |
| MIROC-ESM | 3 | 1 | 1 | r1i1p1 |
| MPI-ESM-LR | 3 | 3 | 3 | r1i1p1, r2i1p1, r3i1p1 |
| MPI-ESM-MR | 3 | 1 | 1 | r1i1p1 |
| MRI-CGCM3 | 4 | 1 | 1 | r1i1p1 |

**Table 1.** CMIP5 GCMs used for deriving the climate perturbations for the PGW simulations.
