# Peer review of "Modeling a tropical-like cyclone in the Mediterranean Sea under present and warmer climate"

_Natural Hazards and Earth System Sciences, 2020_

## Referee Comment (RC1) · Anonymous Referee #1 · 21 Jul 2020

General comments:

This article analyzes the single Medicane Rolf in a regional climate model under present and pseudo global warming conditions of the midst of the 21st century. The relative effects of a warming atmosphere and a warming ocean were analyzed separately, which has not been done before for a Medicane. The approach is indeed very interesting and appealing. However, the study would yield more reliable results by applying the same methodology for a small ensemble of simulations instead of just the current case study with single simulations for present day and future conditions. In the concluding remarks the authors emphasize several times that adding more cases would make the study more robust, and I would strongly agree with this statement. As RCM simulated storm tracks, especially of short-lived and small storms such as Medicanes, tend to be

hard to simulate for a climate model, I doubt that another RCM realization with slightly perturbed conditions (e.g. started with a time lag of one day) would yield similar results. This is e.g. the case for Figure 6, which shows the Medicane track for pseudo-global warming of just the ocean or atmosphere; I think these tracks might look different for ensemble simulations. Differing Medicane tracks and storm developments were already shown in e.g. Cavicchia and von Storch (2012) for small ensembles of Medicane simulations when no large-scale constraint was used. The study would largely benefit from including a mini-ensemble of several realizations for Rolf or alternatively from including one or two additional Medicanes (whereby I would think a small ensemble for the same Medicane would provide more robust results) if the results would be similar.

The advantages of the PWG method and the method itself should be described in more detail; it seems obvious that the effects of a warmer atmosphere or warmer ocean can be examined with it. But please explain why 'a more direct assessment of impacts of future climate change on an extreme weather event' can be achieved with it in comparison to directly using GCM scenarios as forcing data for the RCM.

PGW all is remarkably similar to PBS simulation, there seems to be not much of a climate change effect. Is this statement corresponding to other results in literature? Please put this result better into context of existing articles.

The English is generally o.k., but it could be improved for some parts and formulations.

Due to my concerns with reproducibility of the results for ensemble simulations or other Medicane cases, I suggest major revisions before accepting this article.

Specific comments:

Abstract: What is missing here is the novelty of the work. This is stated later in the concluding remarks: 'The main novelty of this work is the investigation of the relative roles of the atmosphere and ocean, respectively in the medicane's response to projected global warming.' In section 2.2 the following novelty is stated: 'To our knowledge, the

present study is the first investigation to employ the PGW method to a tropical-like cyclone in the Mediterranean Sea.' These novelties should be stated more clearly in the abstract.

Line 107: Do you have a reference for the potential vorticity anomaly statement?

Line 111: Which discrepancies? The importance of air-sea interactions does not exclude the former process, does it?

Line 200: Please state here that PRS stands for present, and not just in the concluding remarks.

Line 214: Why were these periods chosen? Why not e.g. WMO standard periods? Please explain.

Line 216: What happens to greenhouse gases and aerosols in the PWG experiments? Were they changed as well or just the variables described? If not, why weren't they changed and what would the effect be if they were changed as well?

Line 245: Please explain all variables of the equation. The equation is numbered (3), but there is no equation (2). Are all equations starting with equation (3) wrongly numbered?

Line 250 and following: Equation (2) is missing

Line 270: The cyclone tracking method used was not described at all. Please add a subsection the section 2 to describe it in more detail.

Line 507: Wouldn't one expect a warmer atmosphere to inhibit Medicane formation and warmer SSTs to reinforce it? Warmer SSTs would increase the temperature difference between the surface and cold, higher atmospheric layers and thus lead to a decrease in stability, while a warmer atmosphere would increase stability and thus lead to the very effect that was described.

Line 568: What is 'middle future climate adapting PGW technique'? Please rewrite

and explain. I suppose meant is something like applying the PWG method for climate change scenarios according to the middle of the 21st century?

Line 572: The term 'best track' is to my understanding reserved for a quality-checked product provided by different weather services for (mostly tropical) cyclones which was derived by a multitude of analysis and measurement data (such as radar and satellite data), and not just a track derived by some kind of tracking algorithm of reanalysis data. Better term it 'reference track' if that's what was meant.

Technical corrections:

Line 55: assesses instead of assess

Line 59: Insert 'an' in front of initial and 'a' in front of tropical

Line 112: blank is missing after 'on'

Line 125: insert 'to' in front of change

Line 149: southern France

Line 185: Tiedtke misspelled

Line 229: Include 'by' before approximately

Line 251: Add 'a' in front of deep

Line 255: Insert a blank after 50

Line 258: Add 'the' after assess

Line 260: hours

Line 262: Add 'that' after low

Line 269: I would rather use 'simulates' instead of 'observes' for ERA5.

Line 288: 'still a' instead if 'a still'

Line 350: southern France

Line 388: Replace 'double' with 'twice'

Line 401: small what?

Line 422: Insert box after grid

Line 427: through most of the lifecycle

Line 437: defined

Line 455: What does 'in a moderate value of the space' mean? Please rewrite.

Line 456: Insert ', see' in front of Fig.

Line 459: Delete 'to'

Line 462: smaller

Line 466: Delete 'to'

Line 498: 'a much larger' instead of 'much a larger'

Line 519: Insert 'the' in front of increase

Line 577: Insert 'in' in front of ERA5.

Line 579: impact instead of impacts

Line 580: replace 'march over' with 'move into a'

Line 595: 'case studies' instead of 'study cases'

Line 598: medicanes instead of medicane; delete 'the' in front of global

Line 611: Insert 'the' in front of medicane

Line 613: roles instead of role

Figures:

[Figure]

Figure 1: Remove comma in front of Rolf Are the years correct or should they read 2036-2065 and 1976-2005?

Figure 2: Are the years correct or should they read 2036-2065 and 1976-2005? And what domain is shown here? It is bigger than WRF 2nd domain and smaller than the first one. I would suggest showing results for the 1st domain for a) and b). Please add the information that relative humidity is shown in gray and temperature in black for c).

Figure 3: 'ends at' instead of 'is until' Southern France

Figure 5: grid box 'scales' instead of 'labels'

Figure 7: grid box values within 250 km radius

References:

L. Cavicchia and H. von Storch, "The simulation of medicanes in a high-resolution regional climate model," Climate Dynamics, vol. 39, no. 9, pp. 2273–2290, 2012.
* * *

---

## Referee Comment (RC2) · Emmanouil Flaounas (Referee) · 21 Jul 2020

Review of the article "Modeling a tropical-like cyclone in the Mediterranean Sea under present and warmer climate" by Koseki et al.

I read the article with great interest and I found the methods and the topic timely and important. I believe that the paper is adding to our understanding of medicanes under climate change and I support the idea of being eventually published in NHESS. Nevertheless, I have several major concerns about the content, the presentation and interpretation of the results. I hope that several of my comments below will be helpful to improve the paper.

Major comments 1) My first major concern is on the definition and interpretation of

the results. Throughout the introduction it is given the impression that medicanes are sharing same dynamics with tropical cyclones. However, this is not the case at least for the majority of cases. Therefore, I strongly suggest to the authors to revise especially the introduction as well as other parts of the paper, taking into account the following comments:

Lines 90-92: Please note that the detection of cyclones through a cloudless "eye" is a phenomenological criterion and lacks of physical content. Up to now all well known medicane cases are only defined using this subjective, arbitrary criterion. Physical criteria have been used earlier, e.g. by Cavicchia et al. (2013) and recently by Zhang et al. (2020). Nevertheless these criteria include Hart diagrams, wind speed and pressure gradients and thus they are highly dependent on the dataset properties (e.g. resolution as it was stressed by Gaertner et al., 2018). At this point, I strongly suggest to discuss the lack of physical content in the definition of medicanes (please also refer to next comments).

Lines 103-106: An intrusion of trough-like systems or cut-offs over the Mediterranean is a typical event that precedes the formation of medicanes. This is also mentioned in the cited publications of Fita et al., 2006 and Chaboureau et al., 2012 (line 106), but also the more recent ones of Bouin and Lebeaupin Brossier (2020) and Fita and Flaounas (2018). Consequently, medicanes are subject to a baroclinic forcing as other extratropical cyclones. This is also discussed in the results of Fita et al., (2006) and Chaboureau et al., (2012). In fact, the formation of medicanes is not expected to be different from other intense Mediterranean cyclones (Flaounas et al., 2015). This is an important difference from tropical cyclones along with the SST difference from the empirical threshold of 26C (as correctly stressed in lines 97-102). Both of these differences should be discussed along with the fact that there is no physical criterion to qualify a Mediterranean cyclone into a tropical-like system.

Line 109. Please note that Fita and Flaounas (2018) show that deep convection took place while the cyclone was asymmetric and cold core. Moreover, the mature stage

of the cyclone coincided with absence of deep convection or at least with weaker convection than in its initial stages (i.e. during cyclogenesis, when it was a "cold core" system).

Lines 109-114. Please revise this part. Miglietta and Rotunno (2019) show that air-sea interactions are important for the development of only one out of the two analysed medicanes. Similar results were also reached by Carrió et al., (2017) for another case of medicane. In fact, Miglietta and Rotunno (2019) discuss that out of three "kinds" of mechanisms for the formation of medicanes, only one is related to WISHE.

Line 132: I believe that Cavicchia et al., (2014) performed their analysis using a simulation of 10 km of resolution. /if so, please revise.

Line 149: Is it possible to acquire additional information from the fact that Rolf is the first cyclone followed by NOAA as a tropical-one in the Mediterranean? Does it mean for instance that no other cyclone or Medicane before Rolf is to be considered as a tropical-one (at least by NOAA)? How many other Mediterranean cyclones were followed by NOAA after Rolf? Is the NOAA's criterion for tracking tropical cyclones also phenomenological (e.g. tracking spiral clouds in satellite pictures), or does it implicate physical criteria?.

Line 147: I strongly suggest to explain in more detail why Rolf was chosen. Actually the cited studies show a very important presence of deep convection in its centre. In addition, Rolf was related to a rather weak upper tropospheric disturbance. This comes in contrast to other medicanes. Rolf is indeed a far "better" candidate to be considered as "tropical-like", (in the sense that Rolf may unlikely be subject to baroclinic forcing and more plausibly it was driven by convection, thus complying with the WISHE mechanism). Such an entry in the text would make a reasonable connection with previous parts of the introduction on the still uncertain physical definition of medicanes, but also with the validity of the interpretation of the results in the context of climate change (see major comment #4).

[Figure]

Lines 241-242: Please note that Fita and Flaounas (2018) show that warm core and axisymmetry may be achieved due to warm seclusion and not due to the development of convection. This suggests that convection or WISHE could not sustain the cyclone on itself, i.e. tropical transition does not apply to that case study. This is also discussed in Miglietta and Rotunno (2019). Please revise.

To summarize, I suggest to explicitly mention that all known medicanes, if not most, are identified using arbitrary, phenomenological criteria such as the observation of a spiral cloud coverage and a cloudless "eye". Many of these known cases, as shown in previous studies, are not sharing similar dynamics with tropical cyclones in the sense that an upper tropospheric forcing is potentially strong. It is thus important to mention why Rolf is different and how representative it is, when compared to other medicanes (or other intense cyclones).

2) My second major comment goes on the use of English. In several parts, language is understandable but in many parts it is quite familiar and its overall level must be improved. Several minor comments below point towards this direction highlighting several awkward phrasings.

3) After reading the paper, my impression is that the size could be substantially reduced. In fact, I strongly suggest a relatively strong editing by reorganising the two main sections. It seems that paragraphs in sections 3 and 4 are each devoted to a single variable. Both of these sections include a rather long and continuous text where the detailed description of the figures is difficult to be retained. In addition, the focus of the results is often alternated between the different experiments and between ERA5 to PRS. I propose to insert more subsections and to provide to these subsections a content which is based on physical mechanisms rather than physical variables. After all, several paragraphs -especially in section 4- tend to point to the same conclusion, but from the point of view of different variables: how and when the medicane tends to attains a more or less tropical-like structure. Finally, I suggest to omit ERA5 throughout section 3. This would make reading more straight forward.

4) My final major comment goes on the interpretation of the results in the context of climate change. Main results show that higher SST drives Rolf to become stronger, while drier atmosphere is weakening the cyclone. However, as shown in previous studies, upper tropospheric disturbances are constantly interacting with medicanes (as it happens for other intense Mediterranean cyclones). These upper tropospheric systems are usually products of wave breaking over the Atlantic and therefore, the future of Mediterranean cyclones strongly depends on large scale circulation. In addition, the Atlantic Ocean functions as a major source of water vapour (Flaounas et al., 2019) for Mediterranean cyclones and this is not taken into consideration here. Indeed, the boundary conditions only prescribe a background value of relative humidity and not whether water vapour transport towards the Mediterranean will be more (or less) significant in future cyclogenesis events. Therefore, I suggest to be more precise that the results may only relate current cyclones with a background forcing of climate change, rather than reflect the future dynamics of medicanes. However, I find it interesting to stress that Rolf seems to be a system that is least affected by large scale circulation. Consequently, understanding the background forcing of climate change on Rolf's development is of crucial interest for other similar medicanes that might occur in the future.

Minor comments: Line 121: misses "et al"

Line 179: "Miglietta"

Lines 260-276. This paragraph is very detailed and the reader's focus is somewhat shared between NOAA, ERA5 and WRF. I guess that WRF's accuracy in reproducing the track is the important message. I suggest you shorten dramatically this section by providing the most important information as supported by the figure.

Lines 282-283: "develops more vertically", awkward phrasing, please rephrase.

Lines 284-285: I am not sure that cyclone phase diagrams are anyhow related to cyclones intensity. Please explain better this part.

Line 291: The terms presented in Fig. 4 are representative of warm/cold advection and thus they are both expected to be very sensitive to models horizontal resolution. I am not sure if the phrase "stronger warm core" has a "solid" physical interpretation, or if observed differences are mostly due to resolution differences. Would it be more fair to say that PRS reproduces Rolf in a way that cyclone phases match accordingly the ones of ERA5?

Lines 277-293: I am not sure if Figures 4a and 4c provide more information than the ones provided by this paragraph.

Line 304-305: What is meant by "development of the cyclone"? For the period of 3 of November and until the 8 of November, the SLP and latent heat in Fig. 5 seem to be correlated in PRS. Shouldn't an increase of latent heat lead to a stronger cyclone due to a stronger convection and therefore to a decrease of SLP as in PGWSST? Does this mean that Rolf is not behaving as a tropical cyclone (i.e. does not comply with the WISHE mechanism) and thus another physical agent is driving its intensification.

Line 315: "huge amount". Is it possible to quantify this result and compare it to values of previous studies of Mediterranean cyclones and/or other cyclone categories? Is it more than normal? Is it comparable to cyclones developing over open oceans.

Line 341: Make landfall.

Lines 329-352. This large part of section 4 is thoroughly descriptive. It could be shortened by presenting directly the most important differences. After all, the track is also described in the previous section.

Line 353: Figure 7a shows...

Line 358: What is meant by "strength of deepening"?

Line 358: If Figure S1 (also for S2) is indeed important for the presentation of the results then please move it to the main article.

Line 360-361: "warmer climate tends to deepen the centre of the medicane". Please relate cyclones intensity with processes. Also this statement is contradictory with the results in Fig. 7a. It is not the deepening rate or minimum SLP that is different, but the gradient of SLP.

Lines 362-363 and 374: Awkward phrasing, please rephrase.

Line 379: This conclusion seems to overgeneralise the situation where a drier atmosphere is weakening a cyclone and a warmer SST is intensifying it. I suggest to rephrase (see also major comment #4).

Line 381: Figure 7b shows...

Line 385: "Correspondingly to the more rapid decay of the cyclone". Awkward phrasing.

line 385 and throughout the manuscript: "much more". Please quantify your results and compare them to other experiments or previous studies.

Line 392: here and elsewhere (e.g. line 414) what is meant by "inactivated"?

Line 398: Maybe it would be better to move the entire presentation of PRS in the previous section?

Lines 399-400: Could you please verify with the model outputs?

Line 407: "amplitude". Please change to amount; "much larger", as previously mentioned quantify your results and put them into context e.g. by comparing with previous studies. You may compare results of 7c with Figure 8 from Flaounas et al. (2019). It seems that the 2.7 mm/h places Rolf indeed as an outlier system when compared to other intense Mediterranean cyclones (maybe this information is also useful for the introduction).

Line 409. It is here (and in other lines, e.g. 442) quite clear that S1 is important for the presentation of the results. I suggest you move it into the manuscript.

[Figure]

Line 413: "along the cyclone track", or "during cyclone lifetime".

Lines 416-417: Familiar language.

Line 421: I suggest you show the 95th or 98th quantile of wind speed of all grid points within the 250 km radius. This is more objective and will also smooth the plot; In the caption of Figure 7d: "250 km".

Line 434-437: This part was difficult to understand, please clarify. Also please rearrange the narrative or the order of figures so that the important conclusions are complete.

Lines 441-442: I am not sure I understand how warm or cold core (i.e. temperature advection in cyclone phase diagrams) is related to intensity. Is there a straight forward relationship between thermal advection and cyclones intensity. Does for instance the same stand for extratropical cyclones?

Lines 443-456: I am not sure I understand this part. Language could certainly be improved.

Line 473-474: How is size defined? Actually, I am not sure that I understand how the size is related to cyclone phase diagrams. Continuing my previous comment, cyclone phase diagrams correspond to a rather simplistic diagnostic about cyclones core being warmer or colder than its surrounding. However, these diagrams are used here to interpret cyclone dynamics and relationship with other variables. I understand that there are underlying mechanisms that force cyclone phases to coincide with e.g. peaks of precipitation. Could you please be more analytical on these mechanisms.

Line 475-476: This is a very arbitrary comment. I suggest to remove it.

Line 477: Please correct caption of Fig. 10 ("maximum")

Line 487: "similar" instead of "identical".

Lines 485 & 496: "Vigorous". Please rephrase; also avoid familiar language throughout

the text. Such phrasings are open to interpretation. Maybe rewording could help in guiding the reader to focus on the figure details that merit more attention and better support the results, "e.g. the areas where precipitation exceeds XX mm is more narrow in PGWSST and perfectly encircles the cyclone centre. On the other hand, in PGW...".

Line 491-492: Phrasing gives the impression that there is only an arbitrary observation.

Line 497: "still survives". This is only a time frame of rainfall spatial distribution. What if in later or later times the rainfall is more symmetric but weaker? (e.g. Fita and Flaounas, 2018).

Lines 499-500: Does this mean that Rolf as in PRS may not be classified as a hurricane? Actually the whole paragraph from 477 to 500 seems to be based on arbitrary observations. This seems more appealing to a discussion section. I would suggest to use parts of the text for discussing earlier paragraphs.

Line 508-509: Awkward phrasing.

Lines 500-526: This part introduces a new variable (CAPE). It seems to be a continuation of the same motive as in previous sections, i.e. every paragraph is devoted to a single variable. In these lines, the text is very descriptive, lacks of quantification of the results and includes many arbitrary observations. In addition, use of English should be improved.

Line 530: Background humidity is identical only in the boundary conditions but not in the centre of the cyclones in the two experiments. Therefore I do not believe that there can be such a straight forward interpretation of the difference between the two experiments.

Line 536: Please remove "feedback".

Lines 527 to 537: You may omit this part. It basically describes the WISHE mechanism.

Line 537: "consumes CAPE more rapidly": This is not shown in the figures. Also I am

not sure that I understand why this "indicated that the WISHE mechanism works more effectively".

Line 545: "inhibit", maybe "reduce"?

Emmanouil Flaounas

Athens, 20 July 2020

References

Bouin, M.-N. and Lebeaupin Brossier, C.: Surface processes in the 7 November 2014 medicane from air–sea coupled high-resolution numerical modelling, Atmos. Chem. Phys., 20, 6861–6881, https://doi.org/10.5194/acp-20-6861-2020, 2020.

Carrió, D. S., Homar, V., Jansa, A., Romero, R., & Picornell, M. A. (2017). Tropicalization process of the 7 November 2014 Mediterranean cyclone: Numerical sensitivity study. Atmospheric Research, 197, 300-312.

Flaounas, Emmanouil, Shira Raveh-Rubin, Heini Wernli, Philippe Drobinski, and Sophie Bastin. "The dynamical structure of intense Mediterranean cyclones." Climate Dynamics 44, no. 9-10 (2015): 2411-2427.

E Flaounas, L Fita, K Lagouvardos, V Kotroni, Heavy rainfall in Mediterranean cyclones, Part II: Water budget, precipitation efficiency and remote water sources, Climate Dynamics 53 (5-6), 2539-2555

Zhang et al., 2020 Examining the Precipitation Associated with Medicanes in the High-Resolution ERA-5 Reanalysis Data

---

## Author Comment (AC1) · 24 Sep 2020

**Reply to Reviewer#1**
**by**
Koseki, S., Mooney, P. A ., Cabos, W., Gaertner, M. A.,
de la Vara, A., González-Aléman, J.-J.,

We are very grateful to the reviewer for the insightful comments and suggestions. Our responses are coloured **in blue** for the purposes of clarity. Also please note that any corrections/revision corresponding to the reviewer's comments in the revised manuscript are shown in **blue colour**.

Since we have re-performed the simulation with 6 physical ensemble members and all plots have been also re-made. Therefore, we have re-written some descriptions on the figures. Please note that these rewritings are shown in **red colour** in the revised manuscript.

Additionally, following the reviewer#2's comments, we removed most of the descriptions and figures of ERA5 (some of them are transferred to supplemental information).

**General comments:**
**This article analyzes the single Medicane Rolf in a regional climate model under present and pseudo global warming conditions of the midst of the 21st century. The relative effects of a warming atmosphere and a warming ocean were analyzed separately, which has not been done before for a Medicane. The approach is indeed very interesting and appealing. However, the study would yield more reliable results by applying the same methodology for a small ensemble of simulations instead of just the current case study with single simulations for present day and future conditions. In the concluding remarks the authors emphasize several times that adding more cases would make the study more robust, and I would strongly agree with this statement. As RCM simulated storm tracks, especially of short-lived and small storms such as Medicanes, tend to be hard to simulate for a climate model, I doubt that another RCM realization with slightly perturbed conditions (e.g. started with a time lag of one day) would yield similar results. This is e.g. the case for Figure 6, which shows the Medicane track for pseudo-global warming of just the ocean or atmosphere; I think these tracks might look different for ensemble simulations. Differing Medicane tracks and storm developments were already shown in e.g. Cavicchia and von Storch (2012) for small ensembles of Medicane simulations when no large-scale constraint was used. The study would largely benefit from including a mini-ensemble of several realizations for Rolf or alternatively from including one or two additional Medicanes (whereby I would think a small ensemble for the same Medicane would provide more robust results) if the results would be similar.**

Thank you so much for the very constructive comment. We agree that the robustness of our results can be improved. Therefore, we have added 20 more simulations. For each PRS, $PGW_{ALL}$, $PGW_{SST}$, $PGW_{ATMS}$, we have added 5 more simulations that use different combinations of physical schemes (3 different microphysical schemes and 2 different PBL schemes) to the original simulations of medicane Rolf. The combinations and abbreviations of the simulations are given in Table 2. That is, PRS/PGWs have 6 ensemble members in the revised results and we performed the same analyses with the ensemble mean.

In principle, the ensemble-mean results of changes in features of the simulated medicane are approximately consistent with the original results (please see all the figures). Therefore, we conclude that our previous discussions and arguments on the climate change impacts on the medicane are the same, but now more robustness.

The addition of these new simulations has led to a substantial text revision in the abstract and especially in Section 2.1. Please see lines 58 and 202-219.

**The advantages of the PWG method and the method itself should be described in more detail; it seems obvious that the effects of a warmer atmosphere or warmer ocean can be examined with it. But please explain why 'a more direct assessment of impacts of future climate change on an extreme weather event' can be achieved with it in comparison to directly using GCM scenarios as forcing data for the RCM.**

Agreed. Additional information has been added to the section describing the PGW method and the advantages over that of a single RCM realisation. Please see lines 240-244.

**PGW all is remarkably similar to PBS simulation, there seems to be not much of a climate change effect. Is this statement corresponding to other results in literature? Please put this result better into context of existing articles.**

In terms of SLP minimization, PRS and $PGW_{ALL}$ could have very similar results, indicating that the climate change delta has only a minor influence on the medicane. On the other hand, precipitation is enhanced in $PGW_{ALL}$ (e.g., Figs. 6 and Fig.10b) due to enhanced latent heat flux and CAPE. In spite of the similar SLP reductions, the SLP deepening (SLP gradient) is relatively stronger in $PGW_{ALL}$ than in PRS. Therefore, we have concluded that the medicane would be changed modestly by climate change with greater potential for flooding from the increased precipitation. Actually, a similar suggestion of modest intensification is made by Cavicchia et al. (2014) and González-Aléman et al. (2019). We have cited this paper (please see lines 137-145, 510-511, and 615-618).

These results are in general quite similar in the 6 ensemble simulations presented in the revised. Our results are now therefore more robust.

**The English is generally o.k., but it could be improved for some parts and formulations.**

We re-read the manuscript carefully and improved the English writing where required.

**Due to my concerns with reproducibility of the results for ensemble simulations or other Medicane cases, I suggest major revisions before accepting this article.**

As we state above, we added 5 more ensemble simulations with different combinations of physics schemes. Basically, the new simulations produce results which are similar to the original ones. Therefore, we would think that this revision improved the robustness of our discussions.

**Specific comments:**

**Abstract: What is missing here is the novelty of the work. This is stated later in the concluding remarks: 'The main novelty of this work is the investigation of the relative roles of the atmosphere and ocean, respectively in the medicane's response to projected global warming.' In section 2.2 the following novelty is stated: 'To our knowledge, the present study is the first investigation to employ the PGW method to a tropical-like cyclone in the Mediterranean Sea.' These novelties should be stated more clearly in the abstract.**

Thank you for the suggestions. We added the novelty of this work in the abstract. Please see lines 62-63 and 80-82.

**Line 107: Do you have a reference for the potential vorticity anomaly statement?**

We added Miglietta et al. (2016). Please see line 120-121.

**Line 111: Which discrepancies? The importance of air-sea interactions does not exclude the former process, does it?**

Yes. Here, we explained that air-sea fluxes are important for some medicanes but not all of them. We added "importance of the air-sea interaction". Please see line 127.

**Line 200: Please state here that PRS stands for present, and not just in the concluding remarks.**

Following this suggestion, this has been added. Please see lines 224-225.

**Line 214: Why were these periods chosen? Why not e.g. WMO standard periods? Please explain.**

In the revised version of the manuscript, the following text has been added to address this question: *"These periods were chosen on the basis of data availability for CMIP5 CGCMs and to represent 1.5C global warming in the middle of this century."* Please see lines 240-241.

**Line 216: What happens to greenhouse gases and aerosols in the PWG experiments? Were they changed as well or just the variables described? If not, why weren't they changed and what would the effect be if they were changed as well?**

In the WRF experiments, we used the same $CO_2$ and aerosol emissions. The CGCMs computation is driven by such increments of $CO_2$ and aerosol emission. Their influences are included indirectly through the boundary condition for PGW simulations. Especially, if $CO_2$ is also increased in the WRF experiments, the additional global warming effect works and the results would be changed. But, this

could overestimate the impacts of global warming due to double-counting of $CO_2$ effect (indirect and direct).

**Line 245: Please explain all variables of the equation. The equation is numbered (3), but there is no equation (2). Are all equations starting with equation (3) wrongly numbered?**

We added the explanation on the variables. Please see lines 248-255.

**Line 250 and following: Equation (2) is missing**

Added. Please see line 252.

**Line 270: The cyclone tracking method used was not described at all. Please add a subsection the section 2 to describe it in more detail.**

The method of cyclone track is based on minimum sea level pressure. We added this explanation in the revised version of the manuscript. Please see lines 187-189.

**Line 507: Wouldn't one expect a warmer atmosphere to inhibit Medicane formation and warmer SSTs to reinforce it? Warmer SSTs would increase the temperature difference between the surface and cold, higher atmospheric layers and thus lead to a decrease in stability, while a warmer atmosphere would increase stability and thus lead to the very effect that was described.**

As the reviewer comments, such situation could be expected. On the other hand, the warmer atmosphere can contain more water vapour (Clausius-Clapeyron's relation) and the energy available for the medicane development is also potentially increased.

However, as some previous studies and CMIP5 data of climate projection used in this study show, the Mediterranean region tends to be more arid and warmer due to climate change. Our results of PGW$_{ATMS}$ reflect such conditions for convective inhibition. In addition, as long as we are concerned, there is no study that investigates the relative role of a warmer atmosphere and ocean on the characteristics of a tropical-like storm. This is pointed out as the novelty of this work in the abstract and conclusion sections. Therefore, we still think the discussion is worth of writing and of interest for the paper.

**Line 568: What is 'middle future climate adapting PGW technique'? Please rewrite and explain. I suppose meant is something like applying the PWG method for climate change scenarios according to the middle of the 21st century?**

Agreed. **This text has now been changed to** "*future warming conditions of 1.5$^oC$ by applying the PWG method for RCP8.5 according to the middle of the 21st century*". Please see lines 578-579.

**Line 572: The term 'best track' is to my understanding reserved for a quality-checked product provided by different weather services for (mostly tropical) cyclones which was derived by a multitude of analysis and measurement data**

**(such as radar and satellite data), and not just a track derived by some kind of tracking algorithm of reanalysis data. Better term it 'reference track' if that's what was meant.**

Thank you for your suggestion. We changed the word to "reference track". The first definition of reference track is given in Section 2. Please see line 187-189.

**Technical corrections:**
**Line 55: assesses instead of assess**

Corrected.

**Line 59: Insert 'an' in front of initial and 'a' in front of tropical**

Inserted.

**Line 112: blank is missing after 'on'**

Corrected.

**Line 125: insert 'to' in front of change**

Added.

**Line 149: southern France**

Corrected.

**Line 185: Tiedtke misspelled**

Corrected.

**Line 229: Include 'by' before approximately**

Added.

**Line 251: Add 'a' in front of deep**

Added.

**Line 255: Insert a blank after 50**

Inserted.

**Line 258: Add 'the' after assess**

Added.

**Line 260: hours**

Corrected.

**Line 262: Add 'that' after low**

Added.

**Line 269: I would rather use 'simulates' instead of 'observes' for ERA5.**

Corrected.

**Line 288: 'still a' instead if 'a still'**

Corrected.

**Line 350: southern France**

Corrected.

**Line 388: Replace 'double' with 'twice'**

Replaced.

**Line 401: small what?**

Added "amount".

**Line 422: Insert box after grid**

Inserted.

**Line 427: through most of the lifecycle**

Corrected.

**Line 437: defined**

Corrected.

**Line 455: What does 'in a moderate value of the space' mean? Please rewrite.**

We rewrote the sentence to "*the maximum value of the deep warm core phase is smaller than those of PRS and other PGWs.*" Please see line 470-471.

**Line 456: Insert ', see' in front of Fig.**

Added.

**Line 459: Delete 'to'**

Deleted.

**Line 462: smaller**

Corrected.

**Line 466: Delete 'to'**

Deleted.

**Line 498: 'a much larger' instead of 'much a larger'**

Corrected.

**Line 519: Insert 'the' in front of increase**

Inserted.

**Line 577: Insert 'in' in front of ERA5.**

Inserted.

**Line 579: impact instead of impacts**

Corrected.

**Line 580: replace 'march over' with 'move into a'**

Replaced.

**Line 595: 'case studies' instead of 'study cases'**

Corrected.

**Line 598: medicanes instead of medicane; delete 'the' in front of global**

Corrected.

**Line 611: Insert 'the' in front of medicane**

Inserted.

**Line 613: roles instead of role**

Corrected.

**Figures:**
**Figure 1: Remove comma in front of Rolf Are the years correct or should they read 2036-2065 and 1976-2005?**

Deleted. Regarding the year, that is correct.

**Figure 2: Are the years correct or should they read 2036-2065 and 1976-2005? And what domain is shown here? It is bigger than WRF 2nd domain and smaller than the first one. I would suggest showing results for the 1st domain for a) and b). Please add the information that relative humidity is shown in gray and temperature in black for c).**

For (a) and (b), we replaced with the plots of WRF 1$^{st}$ domain. For (c), the vertical profile of temperature and relative humidity at 2011-Nov-05-00UTC has been added for comparison.

**Figure 3: 'ends at' instead of 'is until' Southern France**

Since we changed this figure with 6 ensemble members, the caption has also been modified.

**Figure 5: grid box 'scales' instead of 'labels'**

Corrected. Please note that ERA5 is excluded from this figure following the reviewer#2 comments.

**Figure 7: grid box values within 250 km radius**

Corrected.

**References:**
**L. Cavicchia and H. von Storch, "The simulation of medicanes in a high-resolution regional climate model," Climate Dynamics, vol. 39, no. 9, pp. 2273–2290, 2012.**

Added. Please see lines 153-154.

---

## Author Comment (AC2) · 24 Sep 2020

**Reply to Reviewer#2, Dr. Emmanouli Flaounas**
**by**
Koseki, S., Mooney, P. A ., Cabos, W., Gaertner, M. A.,
de la Vara, A., González-Aléman, J.-J.,

**I read the article with great interest and I found the methods and the topic timely and important. I believe that the paper is adding to our understanding of medicanes under climate change and I support the idea of being eventually published in NHESS. Nevertheless, I have several major concerns about the content, the presentation and interpretation of the results. I hope that several of my comments below will be helpful to improve the paper.**

We are very grateful for the many detailed and constructive comments in your review. We have made every effort to address these helpful comments and we believe that this has greatly improved the manuscript. Our responses to your comments are in blue for clarity.

Following the reviewer#1's comments, we have re-done the simulation wth 6 ensemble members. Consequently all plots have been also re-made and the descriptions on some figures have been re-written. Please note that these rewritings are shown in **red** in the revised version of manuscript.

*Major comments*
**1) My first major concern is on the definition and interpretation of the results. Throughout the introduction it is given the impression that medicanes are sharing same dynamics with tropical cyclones. However, this is not the case at least for the majority of cases. Therefore, I strongly suggest to the authors to revise especially the introduction as well as other parts of the paper, taking into account the following comments:**

*Lines 90-92*: **Please note that the detection of cyclones through a cloudless "eye" is a phenomenological criterion and lacks physical content. Up to now all well known medicane cases are only defined using this subjective, arbitrary criterion. Physical criteria have been used earlier, e.g. by Cavicchia et al. (2013) and recently by Zhang et al. (2020). Nevertheless these criteria include Hart diagrams, wind speed and pressure gradients and thus they are highly dependent on the dataset properties (e.g. resolution as it was stressed by Gaertner et al., 2018). At this point, I strongly suggest to discuss the lack of physical content in the definition of medicanes (please also refer to next comments).**

We have modified and added the following text from lines 90-99:

*"It is well known that severe cyclonic storms occur in the Mediterranean Sea, in particular, from September to March (e.g., Cavicchia et al., 2013). They generate large amount of precipitation and intense winds that severely damage regional economies and infrastructure over the coastal areas in the Mediterranean nations (e.g., Bakkensen, 2017). Although these cyclonic systems have a clear societal importance, methods to robustly detect these cyclones via physical criterion (e.g., Cavicchia et al. (2013) and Zhang et al. (2020)) have remained elusive (Gaertner et al., 2018). The cyclonic systems are typically detected via phenomenological criteria*

*such as the cloud-free "eye". Consequently, most known cyclonic storms in the Mediterranean Sea develop into meso-scale cyclones with a cloud-free "eye" around the cyclone centre, which is also a common feature of tropical cyclones."*

**Lines 103-106: An intrusion of trough-like systems or cut-offs over the Mediterranean is a typical event that precedes the formation of medicanes. This is also mentioned in the cited publications of Fita et al., 2006 and Chaboureau et al., 2012 (line 106), but also the more recent ones of Bouin and Lebeaupin Brossier (2020) and Fita and Flaounas (2018). Consequently, medicanes are subject to a baroclinic forcing as other extratropical cyclones. This is also discussed in the results of Fita et al., (2006) and Chaboureau et al., (2012). In fact, the formation of medicanes is not expected to be different from other intense Mediterranean cyclones (Flaounas et al., 2015). This is an important difference from tropical cyclones along with the SST difference from the empirical threshold of 26C (as correctly stressed in lines 97-102). Both of these differences should be discussed along with the fact that there is no physical criterion to qualify a Mediterranean cyclone into a tropical-like system.**

We have added the following text from lines 100-118:

*"These tropical-like cyclones are called Mediterranean hurricanes or medicanes (this term is used hereafter). Although there are many similarities between medicanes and tropical cyclones, there are also clear differences between them. Firstly, the lifetime of medicanes is shorter than that of most tropical cyclones. Secondly, the development of tropical cyclones generally requires that sea surface temperatures (SSTs) exceed the empirical threshold of 26°C. However, SSTs in the Mediterranean Sea are almost never this warm with autumn and winter SSTs varying from around 18°C to 23°C in the current climate (e.g., Shaltout and Omstedt, 2014; Fig. 2a). This is thus much lower than the empirical threshold of 26°C for tropical cyclone formation and the occurrence of tropical cyclones over such cold SSTs are very rare even in the tropics (cf. Pacific and Atlantic cold tongue, e.g., Jin 1996; Caniaux et al., 2011). Another difference between medicanes and tropical cyclones is that the formation of medicanes is generally preceded by an intrusion of trough-like systems or cut-off lows over the Mediterranean (Fita et al., 2006; Chaboureau et al., 2012; Fita and Flaounas, 2018; Bouin and Lebeaupin Brossier, 2020). In particular, Fita and Flaounas (2018) suggested that some medicanes show hybrid features of tropical and extratropical cyclones, which is more similar to subtropical cyclones (cold core and shallow convection at the mature stage). Consequently, they are subjected to baroclinic forcing like extratropical cyclones (Fita et al., 2006; Chaboureau et al., 2012). As such, it is expected that the formation of medicanes is not different from other intense Mediterranean cyclones (Flauonas et al., 2015), and it should be noted that there is no physical criterion to quantify a Mediterranean cyclone into a tropical-like system."*

**Line 109. Please note that Fita and Flaounas (2018) show that deep convection took place while the cyclone was asymmetric and cold core. Moreover, the mature stage of the cyclone coincided with absence of deep convection or at least with weaker convection than in its initial stages (i.e. during cyclogenesis, when it was a "cold core" system).**

We have added the following text from lines 112-114:

*"In particular, Fita and Flaounas (2018) suggested that some medicanes show hybrid features of tropical and extratropical cyclones, which is more similar to subtropical cyclones (cold core and shallow convection at the mature stage)."*

**Lines 109-114. Please revise this part. Miglietta and Rotunno (2019) show that airsea interactions are important for the development of only one out of the two analysed medicanes. Similar results were also reached by Carrió et al., (2017) for another case of medicane. In fact, Miglietta and Rotunno (2019) discuss that out of three "kinds" of mechanisms for the formation of medicanes, only one is related to WISHE.**

Thank you so much for pointing it out. We have revised the texts from lines 123-130:

*"While Miglietta and Rotunno (2019) showed the importance of air-sea interactions for one medicane out of the two studied, they also suggested that the other case medicane is maintained mainly by mid-latitude baroclinic environment (air-sea fluxes and latent heat flux still help to develop the medicane). This aspect is also suggested by Carrió et al. (2017). These discrepancies on the importance of air-sea interaction in the literature may arise from the dependency of the various works on individual case studies. In particular, the importance of air-sea fluxes can be related to wind-induced surface heat exchange (WISHE) mechanism similar to tropical cyclones (e.g., Emanuel, 1986; Miglietta and Rotunno, 2019)."*

*Line 132***: I believe that Cavicchia et al., (2014) performed their analysis using a simulation of 10 km of resolution. /if so, please revise.**

It is correct that Cavicchia et al (2014) used 10 km grid spacings. However, the text (previously on line 132) refers to global modelling studies while Cavicchia et al. (2014) use a regional climate model. Consequently, we have not revised the statement about the limitations of Coupled GCM simulations arising from their grid spacings. So, we added Cavicchia et al. (2014) as an example of the study of medicanes with a high-resolution regional climate model. Please see lines 153-154.

**Line 149: Is it possible to acquire additional information from the fact that Rolf is the first cyclone followed by NOAA as a tropical-one in the Mediterranean? Does it mean for instance that no other cyclone or Medicane before Rolf is to be considered as a tropical-one (at least by NOAA)? How many other Mediterranean cyclones were followed by NOAA after Rolf? Is the NOAA's criterion for tracking tropical cyclones also phenomenological (e.g. tracking spiral clouds in satellite pictures), or does it implicate physical criteria?.**

We re-consider this part and decided to remove the description on it. Instead, we added more scientific details on Rolf as in response to the other comment. Please see lines 163-174.

**Line 147: I strongly suggest to explain in more detail why Rolf was chosen. Actually the cited studies show a very important presence of deep convection**

in its centre. In addition, Rolf was related to a rather weak upper tropospheric disturbance. This comes in contrast to other medicanes. Rolf is indeed a far "better" candidate to be considered as "tropical-like", (in the sense that Rolf may unlikely be subject to baroclinic forcing and more plausibly it was driven by convection, thus complying with the WISHE mechanism). Such an entry in the text would make a reasonable connection with previous parts of the introduction on the still uncertain physical definition of medicanes, but also with the validity of the interpretation of the results in the context of climate change (see major comment #4).

Our review of the literature suggests that Rolf is one of the more intense medicanes wth long-lasting tropical-like characteristics. Additionally, Rolf occurred around the Balearic Islands where many medicanes are generated. That is our main motivation to investigate Rolf and its future change. We have revised the texts from lines 166-174:

*"Since Rolf was a highly destructive medicane for coastal communities in many Mediterranean countries and is one of the most intense medicanes (e.g., Dafis et al., 2018), it is important to assess how these types of medicanes will respond to climate change in near future. Medicane Rolf generated the deep cumulus convection and persisted with tropical cyclone-like characteristics longer than other Mediterranean storms and vortices (e.g., Miglietta et al., 2013). Moreover, according to Miglietta et al. (2013), Rolf occurred around the Balearic Islands, which is a hot spot of medicane genesis. Therefore, it will be interesting and important to investigate the impacts of climate change on this type of Mediterranean storm."*

Lines 241-242: Please note that Fita and Flaounas (2018) show that warm core and axisymmetry may be achieved due to warm seclusion and not due to the development of convection. This suggests that convection or WISHE could not sustain the cyclone on itself, i.e. tropical transition does not apply to that case study. This is also discussed in Miglietta and Rotunno (2019). Please revise.

We revised that part. Please see lines 274-275.

To summarize, I suggest to explicitly mention that all known medicanes, if not most, are identified using arbitrary, phenomenological criteria such as the observation of a spiral cloud coverage and a cloudless "eye". Many of these known cases, as shown in previous studies, are not sharing similar dynamics with tropical cyclones in the sense that an upper tropospheric forcing is potentially strong. It is thus important to mention why Rolf is different and how representative it is, when compared to other medicanes (or other intense cyclones).

As outlined above, we have included a more detailed discussion on medicanes in response to the comments. According to Miglietta et al. (2013), Rolf is one of the more intense medicanes and it showed a tropical transition. We also added this rationalization for choosing Rolf as our case study. Please see lines 166-174.

2) My second major comment goes on the use of English. In several parts, language is understandable but in many parts it is quite familiar and its overall level must be improved. Several minor comments below point towards this

**direction highlighting several awkward phrasings.**

We have re-read our manuscript more carefully and corrected expressions in the reviewer's minor comments.

**3) After reading the paper, my impression is that the size could be substantially reduced. In fact, I strongly suggest a relatively strong editing by reorganising the two main sections. It seems that paragraphs in sections 3 and 4 are each devoted to a single variable. Both of these sections include a rather long and continuous text where the detailed description of the figures is difficult to be retained. In addition, the focus of the results is often alternated between the different experiments and between ERA5 to PRS. I propose to insert more subsections and to provide to these subsections a content which is based on physical mechanisms rather than physical variables. After all, several paragraphs -especially in section 4- tend to point to the same conclusion, but from the point of view of different variables: how and when the medicane tends to attains a more or less tropical-like structure. Finally, I suggest to omit ERA5 throughout section 3. This would make reading more straight forward.**

Thank you so much for the comment. For the mechanism showing the plots of precipitation, latent heat flux, CAPE, and wind speed, we explain and discuss why Rolf is modified by different background in sections 4, 5 and 6. Since Rolf has more tropical-like features (deep warm core and deep convection), such plots are helpful in order to explain its mechanism of formation and maintenance. This study pays more attention to how the medicane will be changed more than its fundamental dynamics of development since previous studies have shown the dynamics of medicanes. However, in response to the other comment, we moved Fig.S2 (OLR) to Fig. 12 and provide more discussion on the differences in deep cumulus convection associated with simulated Rolf and our simulations in section 5, in combination with the discussion on CAPE. We believe that this revision gives more physical aspects. Please see lines 536-570.

On the other hand, we agree with the fact that our paper is a bit repetitive and the part of ERA5 could be omitted. However, we recognize that some readers may prefer to see the realism of our PRS simulation. Therefore, we have moved the figures of ERA5 to supplemental information and substantially reduced its description.

**4) My final major comment goes on the interpretation of the results in the context of climate change. Main results show that higher SST drives Rolf to become stronger, while drier atmosphere is weakening the cyclone. However, as shown in previous studies, upper tropospheric disturbances are constantly interacting with medicanes (as it happens for other intense Mediterranean cyclones). These upper tropospheric systems are usually products of wave breaking over the Atlantic and therefore, the future of Mediterranean cyclones strongly depends on large scale circulation. In addition, the Atlantic Ocean functions as a major source of water vapour (Flaounas et al., 2019) for Mediterranean cyclones and this is not taken into consideration here. Indeed, the boundary conditions only prescribe a background value of relative humidity and not whether water vapour transport towards the Mediterranean will be more (or less) significant in future cyclogenesis events. Therefore, I suggest to be more precise that the results may only relate current cyclones with a background forcing of climate change, rather than reflect the future dynamics of medicanes. However, I find it interesting to stress that Rolf seems**

**to be a system that is least affected by large scale circulation. Consequently, understanding the background forcing of climate change on Rolf's development is of crucial interest for other similar medicanes that might occur in the future**

We added the following texts to emphasize "background change". Please see lines 619-627.

*"The PGW technique is a powerful tool to investigate the impacts of climate change on the weather systems in the future. However, our results in this paper include only the climate changes in background such as temperature, relative humidity, SST and etc. In this framework, any changes in extratropical dynamics like wave breaking and large-scale circulation as a source of medicanes are not directly considered. Additionally, as Flaounas et al. (2019) suggest, the water vapor transport from the North Atlantic sector will be modified and significantly influences the medicane frequency and intensity. The PGW approach does not reflect directly such future change in water vapour transport. Nonetheless, we can conclude that the background change associated with global warming will have some impact on the medicane development."*

**Minor comments: Line 121: misses "et al"**

**Line 179: "Miglietta"**

Corrected.

**Lines 260-276. This paragraph is very detailed and the reader's focus is somewhat shared between NOAA, ERA5 and WRF. I guess that WRF's accuracy in reproducing the track is the important message. I suggest you shorten dramatically this section by providing the most important information as supported by the figure.**

Following major comment#3, we moved ERA5's figure to supplemental information and shortened section 3 more focusing on WRF's ability to reproduce Rolf. Please see the entire section 3.

**Lines 282-283: "develops more vertically", awkward phrasing, please rephrase.**

Since Figure 4a has been removed (replying to other minor comments), this expression has also been removed.

**Lines 284-285: I am not sure that cyclone phase diagrams are anyhow related to cyclones intensity. Please explain better this part.**

Since Figure 4a has been removed (replying to other minor comments), this expression has also been removed.

**Line 291: The terms presented in Fig. 4 are representative of warm/cold advection and thus they are both expected to be very sensitive to models horizontal resolution. I am not sure if the phrase "stronger warm core" has a "solid" physical interpretation, or if observed differences are mostly due to resolution differences. Would it be more fair to say that PRS reproduces Rolf in a way that cyclone phases match accordingly the ones of ERA5?**

We rephrased it. Please see lines 304-305.

**Lines 277-293: I am not sure if Figures 4a and 4c provide more information than the ones provided by this paragraph.**

We agree. These figures have been removed from the manuscript since Fig.4b and d already provide sufficient information on cyclone phase.

**Line 304-305: What is meant by "development of the cyclone"? For the period of 3 of November and until the 8 of November, the SLP and latent heat in Fig. 5 seem to be correlated in PRS. Shouldn't an increase of latent heat lead to a stronger cyclone due to a stronger convection and therefore to a decrease of SLP as in PGWSST? Does this mean that Rolf is not behaving as a tropical cyclone (i.e. does not comply with the WISHE mechanism) and thus another physical agent is driving its intensification.**

Before the SLP deepening around 00UTC-08, the latent heat flux is the strongest (after 00UTC-07) and precipitation is maximum around 20UTC-07. This could indicate that (1) the cyclone gains more water vapour and that (2) diabatic heating due to condensation provides energy for cyclone development. This is similar to the development of tropical cyclone. Please see new Fig.6.

**Line 315: "huge amount". Is it possible to quantify this result and compare it to values of previous studies of Mediterranean cyclones and/or other cyclone categories? Is it more than normal? Is it comparable to cyclones developing over open oceans.**

According to Miglietta and Rotunno (2018), the latent heat flux of the October, 2006 case is 1800 W/m$^2$ at the cyclone peak. Their other case of December, 2005 has a value of 1000 W/m$^2$ at the cyclone peak. In our case, at the cyclone peak, the value is about 740 w/m$^2$ in the 6-member ensemble.

However, this difference could be due to the geographical location of the cyclone. Their cyclones' centres of October,1996/ December, 2006 locate 38N-39N/34-35N which is more southern than Rolf (November, 2010), our case, medicane Rolf (at the peak, its latitude is 41N-42N). When the cyclone is located more southerly, the dry air can advect from the African continent and evaporation will be enhanced effectively. The underlying SST is also warmer near the African continent (the cases of Miglietta and Rorunno, 2018) than near Europe (our case).

We added this quantification and discussion. Please see lines 320-324.

**Line 341: Make landfall.**

Corrected.

**Lines 329-352. This large part of section 4 is thoroughly descriptive. It could be shortened by presenting directly the most important differences. After all, the track is also described in the previous section.**

Because the plots have been remade in order to include the 6 ensemble simulations, this part has been re-written drastically to describe the remarkable differences among tracks. Please see lines 350-375.

**Line 353: Figure 7a shows...**

Corrected.

**Line 358: What is meant by "strength of deepening"?**

We meant the SLP gradient here. The sentence has been rephrased. Please see lines 382.

**Line 358: If Figure S1 (also for S2) is indeed important for the presentation of the results then please move it to the main article.**

As suggested, we have moved Fig.S1 to Fig.7. However, Fig.S2 is more useful for the discussion on difference in cumulus convection in Section5 in addition to CAPE. This revision is related to the reviewer's major comment#2 and other minor comment below. Please see lines 619-627.

**Line 360-361: "warmer climate tends to deepen the centre of the medicane". Please relate cyclones intensity with processes. Also this statement is contradictory with the results in Fig. 7a. It is not the deepening rate or minimum SLP that is different, but the gradient of SLP.**

Here, we used "deepening" as SLP gradient. But, this terminology is wrong. In the ensemble simulations, SLP is slightly lower in $PGW_{ALL}$ than in PRS, but still almost same (please see new Fig.6a). And, the SLP gradient is much stronger in $PGW_{All}$ than in PRS (please see new Fig. 7).

To avoid misusing the terminology, we rephrased it in lines 383-386.

**Lines 362-363 and 374: Awkward phrasing, please rephrase.**

This statement has been removed from the manuscript as it does not add any value to the presented results.

**Line 379: This conclusion seems to overgeneralise the situation where a drier at- mosphere is weakening a cyclone and a warmer SST is intensifying it. I suggest to rephrase (see also major comment #4).**

As replied to major comment#4, we agreed that our study showed the impacts of changes of the atmospheric and oceanic background associated with global warming on medicane. Therefore, here, we have modified the text to emphasises "background" Please see line 401-402.

**Line 381: Figure 7b shows...**

Corrected.

**Line 385: "Correspondingly to the more rapid decay of the cyclone". Awkward phrasing.**

From the ensemble simulations, the decay rate of latent heat flux in PRS and PGW$_{ALL}$ are almost identical. Therefore, we removed this sentence.

**line 385 and throughout the manuscript: "much more". Please quantify your results and compare them to other experiments or previous studies.**

As stated in the previous comment, we removed the sentence including the expression here. Also we added more quantifications throughout the manuscript.

**Line 392: here and elsewhere (e.g. line 414) what is meant by "inactivated"?**

We have replaced this word here and elsewhere in the revised version of the manuscript by clearer descriptions.

**Line 398: Maybe it would be better to move the entire presentation of PRS in the previous section?**

We agree on this. This part has been moved to the previous section. Also, the description on maximum wind speed in PRS has been also moved to the previous section. We modified Fig.4 by adding precipitation. Please see lines 324-338.

**Lines 399-400: Could you please verify with the model outputs?**

In this study, we do not aim to investigate the fundamental dynamics of Rolf's development and we understand that the mentioned statement was too speculative. Therefore, we changed our description there. However, the initial intense rainfall can still be related to the initial disturbance. Please see lines 327-328. Also, we added a figure of SLP and precipitation (for this reply) averaged from 00UTC to 12UTC on 6th-Nov in PRS's ensemble mean.

[Figure]

Figure R1. SLP and precipitation averaged from 00UTC to 12UTC on 6, Nov in PRS ensemble.

**Line 407: "amplitude". Please change to amount; "much larger", as previously mentioned quantify your results and put them into context e.g. by comparing with previous studies. You may compare results of 7c with Figure 8 from Flaounas et al. (2019). It seems that the 2.7 mm/h places Rolf indeed as an outlier system when compared to other intense Mediterranean cyclones (maybe this information is also useful for the introduction).**

Corrected. Thank you so much for the useful suggestion. That point is very interesting. Actually, 2.7mm/h (in ensemble, 2.6mm/h) is for PGW$_{SST}$ and PRS has 1.5mm/h. These values are equivalent to 31.2mm/12h and 18mm/12h, respectively. Both are located in the "intense" spots shown in Fig.8 of Flaounas et al. (2019). We added a brief discussion on this. Please see lines 428-432.

**Line 409. It is here (and in other lines, e.g. 442) quite clear that S1 is important for the presentation of the results. I suggest you move it into the manuscript.**

We moved S1 to Fig.7 in the revised version of the manuscript.

**Line 413: "along the cyclone track", or "during cyclone lifetime".**

Corrected.

**Lines 416-417: Familiar language.**

Corrected.

**Line 421: I suggest you show the 95th or 98th quantile of wind speed of all grid points within the 250 km radius. This is more objective and will also smooth the plot; In the caption of Figure 7d: "250 km".**

Thank you so much for the suggestion. We re-plotted that figure by showing averaged winds exceeding 95$^{th}$ percentile of hourly data in each simulation. Please see new version of Fig. 6d.

**Line 434-437: This part was difficult to understand, please clarify. Also please rearrange the narrative or the order of figures so that the important conclusions are complete.**

This paragraph has been rephrased:

*"In PGW$_{ATMS}$, during 06 and 07-Nov, the MWS is stronger than that in PRS. However, after 0000UTC on 08-Nov, the MWS in PGW$_{ATMS}$ is weaker than that in PRS resulting in a smaller maximum amplitude of MWS during the cyclone tracking in PGW$_{ATMS}$ is smaller than in PRS (21m/s for PGW$_{ATMS}$ and 24m/s for PRS). In addition, as seen in Fig. S2, the ratio of grid boxes with weaker wind speeds (category of 5 to 10m/s) is larger in PGW$_{ATMS}$ than in PRS (in particular, 12UTC-07 and 08UTC-08). That is, the area of strong winds is much smaller in PGW$_{ATMS}$ than in PRS (the horizontal distribution of winds will be given in Fig. 9)."*

Please see lines 448-455.

For this revision, we added new Fig.S2 showing a probability density function of wind speeds in PRS and PGW$_{ATMS}$.

**Lines 441-442: I am not sure I understand how warm or cold core (i.e. temperature advection in cyclone phase diagrams) is related to intensity. Is there a straight forward relationship between thermal advection and cyclones intensity. Does for instance the same stand for extratropical cyclones?**

Our argument was too speculative without any concrete evidence. Therefore, we removed that sentence from the manuscript.

**Lines 443-456: I am not sure I understand this part. Language could certainly be improved.**

In the original manuscript, we described each cyclone phase space in details. However, in the revised manuscript, we show 6 lines in each simulation (please see new Fig.8). Thus, we avoid describing unnecessary details and focus more on the overview of differences among the simulations. Please see lines 456-472.

**Line 473-474: How is size defined? Actually, I am not sure that I understand how the size is related to cyclone phase diagrams. Continuing my previous comment, cyclone phase diagrams correspond to a rather simplistic diagnostic about cyclones core being warmer or colder than its surrounding. However, these diagrams are used here to interpret cyclone dynamics and relationship with other variables. I understand that there are underlying mechanisms that force cyclone phases to coincide with e.g. peaks of precipitation. Could you please be more analytical on these mechanisms.**

Regarding the size mentioned here, we meant that the radius of maximum wind speed is smaller in $PGW_{ATMS}$ than in other simulations. In response to an earlier comment, we added new Fig.S2 showing histograms of wind speeds. We rephrased the text with New Fig. S2. Please see lines 452-455 and 485-487.

Regarding the relationship between cyclone phase space and intensity, as replied to other comment above, our argument was too speculative and sufficient evidence. Therefore, we removed that sentence from the manuscript.

**Line 475-476: This is a very arbitrary comment. I suggest to remove it.**

Removed.

**Line 477: Please correct caption of Fig. 10 ("maximum")**

Corrected.

**Line 487: "similar" instead of "identical".**

Done.

**Lines 485 & 496: "Vigorous". Please rephrase; also avoid familiar language throughout the text. Such phrasings are open to interpretation. Maybe rewording could help in guiding the reader to focus on the figure details that merit more attention and better support the results, "e.g. the areas where precipitation exceeds XX mm is more narrow in PGWSST and perfectly encircles the cyclone centre. On the other hand, in PGW...".**

We re-read again carefully and familiar phrasings have been corrected.

**Line 491-492: Phrasing gives the impression that there is only an arbitrary observation.**

This sentence has been removed.

**Line 497: "still survives". This is only a time frame of rainfall spatial distribution. What if in later or later times the rainfall is more symmetric but weaker? (e.g. Fita and Flaounas, 2018).**

We agree. This statement has been removed.

**Lines 499-500: Does this mean that Rolf as in PRS may not be classified as a hurri- cane? Actually the whole paragraph from 477 to 500 seems to be based on arbitrary observations. This seems more appealing to a discussion section. I would suggest to use parts of the text for discussing earlier paragraphs.**

This statement has been removed from this section and we moved Fig.S2 to Fig.12 (OLR in each simulation) to section 5 in order to discuss more details on differences in the simulated medicane. Please see lines 536-562.

**Line 508-509: Awkward phrasing.**

Rewritten to: "*In this section, we discuss the roles of the atmosphere and the ocean in the medicane's response to future warming*"

Please see lines 514-515.

**Lines 500-526: This part introduces a new variable (CAPE). It seems to be a continuation of the same motive as in previous sections, i.e. every paragraph is devoted to a single variable. In these lines, the text is very descriptive, lacks of quantification of the results and includes many arbitrary observations. In addition, use of English should be improved.**

We added a new OLR figure (Fig.12) to this section to provide a more insightful discussion on the differences among medicanes in each simulation. In particular, OLR is a reasonable indicator for changes in deep cumulus convection associated with this medicane. Please see lines 536-549.

**Line 530: Background humidity is identical only in the boundary conditions but not in the centre of the cyclones in the two experiments. Therefore I do not believe that there can be such a straight forward interpretation of the difference between the two experiments.**

Agreed. This part has been removed. Additionally, we discussed the differences of the simulated medicane between PRS and PGW$_{SST}$. The difference between the two experiments is related to the SST boundary condition. It is obvious that the warmer SST will modulate the humidity field entirely giving more water vapour as our figures show.

**Line 536: Please remove "feedback".**

Removed.

**Lines 527 to 537: You may omit this part. It basically describes the WISHE mechanism.**

Omitted.

**Line 537: "consumes CAPE more rapidly": This is not shown in the figures. Also I am not sure that I understand why this "indicated that the WISHE mechanism works more effectively".**

This sentence has been removed.

**Line 545: "inhibit", maybe "reduce"?**

Corrected.

---

## Author Response (AR2)

**Reply to Reviewer#1**

**General comments:**
**The authors added 20 more simulations to their analysis using different physical schemes. As the individual simulations basically agree with each other the results did become more robust. It is indeed very interesting to see how well the Medicane is reproducible with the different set ups. The PWG method was described in more detail and also the description of Medicane dynamics was improved. The novelty of the article was now stated more clearly in the abstract and conclusions. Therefore my general and specific comments were taken into account.**

**The main point still to be solved is the language of the article. There is a large number of smaller misspellings, missing or wrongly placed punctuation marks, many grammatical errors or sentences which are hardly or not at all understandable. In the former version the English was generally o.k., but in the new version especially the revised parts have to be improved before the article can be published.**

Thank you very much for reviewing our revised manuscript and giving more useful comments on our revised manuscript. Please note that any corrections/revision corresponding to the reviewer's comments in the revised manuscript are shown in blue colour for clarity. We carefully read our manuscript again to improve our English.

Please note that "Ibiza" was replaced with "Menorca". That was geographic error in the previous manuscripts.

**Reply to Reviewer#2 Dr. Emmanouli Flaounas**

**I read the revised version carefully and I found the manuscript to be improved. Most of my comments and suggestions were adequately addressed, although I find that English could be still improved. Nevertheless, the text is understandable and the messages are clear. I only have some minor languages comments on the corrections done by the authors.**

Thank you very much for reviewing our revised manuscript and giving more useful comments on our revised manuscript. Please note that our response and any corrections/revision corresponding to the reviewer's comments in the revised manuscript are shown in blue colour for clarity. We carefully read our manuscript again to improve our English.

Please note that "Ibiza" was replaced with "Menorca". That was geographic error in the previous manuscripts.

**Line 63: "medicanes"**

Corrected.

**Line 65: "delta"? This could become more precise?**

We added (difference between future and present climate)

**Line 67: "most of the medicane characteristics moderately intensify". Either be more precise or remain to a more general remark, e.g. the cyclone is fairly unchanged.**

Yes, that is why we wrote after this sentence like "e.g., wind speed, uprake of water vapour and precipitation". These are intensified characteristics of the medicane.

**Line 68: what is the "maximum depression of sea level pressure (SLP) minimum"?**

Rephrased to minimum sea level pressure (SLP)

**Line 81 "insights into", what is the "anthropogenic warmer ocean"?**

Corrected and removed anthropogenic.

**Line 91: "amounts"**

Corrected.

**Line 93: Please remove "nations".**

Removed.

**Lines 93-99: I suggest to rephrase. A high number of cyclonic storms may occur during a year. However, it is only few of them that qualify as medicanes mainly due to phenomenological criteria such as cloud-free "eye". I suggest to remove Lines 97-99.**

Rephrased and lines 97-99 are removed. Please see lines 93-96.

**Line 101: What similarities are you referring to? If this is the "eye" formation then I suggest to better articulate with the previous paragraph.**

We added "which is just one of characteristics of tropical cyclones in the previous paragraph". Please see lines 95-96.

**Line 106: This temperature range is not homogeneous in the Mediterranean Sea. IT is my impression that 23 degrees is rarely reached in the areas where medicanes are more frequently developed.**

Here, we argue how cold the SST is in the Mediterranean Sea compared to the threshold of tropical cyclone generations over the tropical oceans.

**Line 110: "triggered" instead of "preceded".**

Replaced.

**Line 122: I would say that the case study of Fita and Flaounas (2018) is not relevant to tropical transition. This is indeed contradictory with statement in line 275.**

We removed Fita and Flaounas (2018) from this line.

**Lines 127-130: I do not understand this part.**

We deleted this part from the manuscript.

**Line 131: "...some cases... involves tropical transition". Please provide details and references on the cases where tropical transition was relevant and how TT was involved.**

As a reference of tropical transition, we cited Mazza et al (2017). At line 120, we added "due to the warm seclusion".
Since we realized line131 does not connect with the rest of this paragraph, we rephrased that part in order to have a good flow of description. Please see lines 125.

**Line 170: What tropical cyclone-like characteristics are you referring to?**

We added "deep warm core". Please see line 163.

**Line 189: It seems quite odd to start figures narrative with a reference to supplement material.**

We moved that paragraph after the model setting. Please see lines 217-225.

**Line 312: Here and throughout the manuscript, please be more careful when describing the processes. The development of the cyclone "can be linked" to convection which in turn can be favoured by air-sea exchanges.**

Yes, because this paper's main aim is not on the fundamental mechanism of medicane Rolf, we have not used any strong expressions for mechanism of the medicane.

**Line 316: "Decrease" instead of "reduce".**

Corrected.

**Line 317: Awkward phrasing.**

Rephrased. Please see lines 316-317.

**Line 320-322: I am not sure I understand this phrase. Is by "grid box" meant a grid point? If so, why this value is representative for the cyclone processes if it only refers to a very localised peak?**

Yes, it means a grid point. Here, we would like to compare the maximum value of latent heat flux between our case and other cases of medicanes from the reference. As the reviewer says, it is much better to compare some averaged value like our Figure 6, but it is impossible to have the averaged latent heat flux of other medicanes cases in the reference. Therefore, here, we just showed maximum value of latent heat flux for quick comparison.

**Line 324: "enhance evaporation more effectively". This is awkward phrasing. Is it meant that warmer SSTs may favour evaporation in the other studies, compared to Rolf here?**

Yes. We rephrased it. Please seee line 322-323.

**Lines 333-334: I am not sure that I understand the definition of MWS. Why it is not just the 95th percentile of wind speed of all grid points within an area of 250 km from cyclones centre?**

That is what we estimated the MWS. We missed to describe within a area of 250km from cyclone centre. We added it. Please see line 332.

**Line 336-337: How can two different fields with different units vary consistently? Do you mean that they both increase and decrease in phase?**

Because the MWS grids are concentrated around the medicane centre where the SLP gradient is intense. The stronger SLP gradient, the stronger SLP depression is. Therefore, the MWS and SLP show some coherent variation.

**Lines 427-432: There are multiple repetitions here. Please rephrase to clarify the content.**

We removed the repetitive expression here.

**Line 468: what is meant by "similar"? Is it meant that both take place in the same time?**

Here, we refer to the phase shift of the simulated medicane in $PGW_{ATMS}$ and it is similar to the phase shift in PRS and $PGW_{ALL}$.

**Line 469-472: "development of a deep warm core is not as strong...". What is meant by "strong development"?**

We removed this sentence because of repetitive expression. Please see lines 471-474.

**Line 474: Perhaps "present climate" instead of "present day"?**

Replaced.

**Line 487: Either be more precise, or remove this phrase so that the S2 is not necessary for understanding the content.**

We removed the sentence.

**Line 550: what is meant by "activated efficiently"?**

We changed it to "enhanced".

**Line 622: I am not sure what is meant by "source of medicanes".**

We changed it to "that is an initial disturbance for medicanes". Please line 623-624.

**Line 627: Please be more precise on "some impact". Is it fair to say that the background has a limited impact on the development of the medicane?**

Here, we would like to conclude that the background has not large, but certain influence on the medicane as we showed the response of wind and precipitation. Therefore, we changed "some" to "moderate". Please see line 628.

---

## Author Response (AR3)

To the Editor of *Natural Hazards and Earth System Sciences*
Dear Prof. Dr. Joaquim G. Pinto

      Thank you very much for evaluating our revised manuscript and decision. We are very happy that our manuscript has been accepted with correction. We corrected the suggested mistyping and please note that the corrections are shown in red color.
      Again, we really appreciate for the editor's kind consideration, evaluation and making decision throughout the reviewing process of NHESS.

Sincerely,
Shunya Koseki

Shunya.Koseki@gfi.uib.no
+47 55 58 98 24
Geophysical Institute,
University of Bergen,
Allegaten 70, Bergen, Norway